# SND3 is the membrane insertase within a distinct SEC61 translocon complex

Tzu-Jing Yang [1], Saumyak Mukherjee [2], Julian D. Langer [3,4], Gerhard Hummer [2,5] & Melanie A. McDowell [1]✉

During the biogenesis of most eukaryotic integral membrane proteins (IMPs), transmembrane domains are inserted into the endoplasmic reticulum membrane by a dedicated insertase or the SEC61 translocon. The SRP-independent (SND) pathway is the least understood route into the membrane, despite catering for a broad range of IMP types. Here, we show that *Chaetomium thermophilum* SND3 is a membrane insertase with an atypical fold. We further present a cryo-electron microscopy structure of a ribosome-associated SND3 translocon complex involved in co-translational IMP insertion. The structure reveals that the SND3 translocon additionally comprises the complete SEC61 translocon, CCDC47 and TRAPα. Here, the SEC61β N-terminus works together with CCDC47 to prevent substrate access to the translocon. Instead, molecular dynamics simulations show that SND3 disrupts the lipid bilayer to promote IMP insertion via its membrane-embedded hydrophilic groove. Structural and sequence comparisons indicate that the SND3 translocon is a distinct multipass translocon in fungi, euglenozoan parasites and other eukaryotic taxa.

Integral membrane proteins (IMPs) represent 20–30% of all proteomes and perform essential biological processes, including membrane transport, cellular signalling and biosynthesis. The biogenesis of the majority of eukaryotic IMPs starts with their insertion into the endoplasmic reticulum (ER) membrane. During this process, hydrophobic transmembrane domains (TMDs) emerging from the ribosome are first recognised and shielded by targeting factors, before their faithful integration into the membrane by dedicated insertases or the SEC61 translocon. As IMPs are hugely divergent in the position, number and biochemical properties of their TMDs, the cell has evolved distinct pathways to cater for the targeting and insertion of different types of IMP substrates. As a general principle, these pathways utilise either co-translational targeting by the signal recognition particle (SRP) or post-translational targeting by routes such as the guided entry of tail-anchored proteins (GET) or the ER membrane protein complex (EMC) pathways[1,2].

Besides these three well-characterised pathways, evidence for a novel SRP-independent (SND) pathway was found in *Saccharomyces cerevisiae* and humans[3,4]. Three protein components have been implicated in the yeast SND pathway, namely SND1, SND2 and SND3[3]. Cytosolic SND1 has been found associated with ribosomes[5] and is consequently hypothesised to be a targeting factor in the pathway, whilst the ER membrane proteins SND2 and SND3 are expected to form a heterodimeric complex that could act as a membrane receptor, insertase and/or chaperone[3]. Recently, the human homologue of SND2, TMEM208, was shown to be a receptor for the targeting factor SRP and to accelerate the release of IMP substrates from its substrate-binding M domain, indicating the SND pathway is to some extent SRP-dependent[6]. In addition, predicted structures for SND2 and SND3 indicate that both proteins have a significant number of hydrophilic residues within the TMDs[7], a characteristic shared with membrane insertases like those of the Oxa1 superfamily[8]. However, as SND2 and SND3 have also been found associated with the SEC61 translocon, the

[1]Membrane Protein Biogenesis Research Group, Max Planck Institute of Biophysics, Frankfurt am Main, Germany. [2]Department of Theoretical Biophysics, Max Planck Institute of Biophysics, Frankfurt am Main, Germany. [3]Membrane Proteomics and Mass Spectrometry, Max Planck Institute of Biophysics, Frankfurt am Main, Germany. [4]Mass Spectrometry, Max Planck Institute for Brain Research, Frankfurt am Main, Germany. [5]Institute for Biophysics, Goethe University Frankfurt, Frankfurt am Main, Germany. ✉e-mail: melanie.mcdowell@biophys.mpg.de

central hub for protein import at the ER membrane, their relative contribution to insertion and folding of substrates is currently unclear[3,4].

Data derived from genetic depletion of SND components suggest that the SND pathway can cater for substrates not efficiently recognised by the other pathways, specifically IMPs with their first TMD in the middle of the protein sequence and SRP-independent substrates such as glycosylphosphatidylinositol (GPI) anchored proteins and short secretory proteins[3,9,10]. In addition, the SND pathway was shown to serve as a 'back-up' pathway to deliver SRP and GET substrates when these canonical pathways are impaired[3,4], suggesting IMPs with their first TMD positioned at the N- or C-terminus can also be accommodated. Altogether, this indicates the SND pathway has a remarkably broad clientele and a lack of substrate specificity based solely on the position of the first TMD. Indeed, more recent studies focused on TMEM208 strongly indicate an involvement in multipass IMP biogenesis[6,11,12], suggesting the number of TMDs in a given substrate protein could alternatively determine its specificity for the SND pathway.

Given the diversity of proposed SND substrates, it is also fundamentally unknown whether the pathway proceeds via co- or post-translational membrane insertion mechanisms. Indeed, the involvement of SRP as a targeting factor would imply a co-translational route[6], yet some putative substrates require auxiliary SEC61 translocon components that prevent ribosome-association with the translocon and instead mediate post-translational translocation[3]. It is also intriguing that SND2 is the only component of the pathway with strict sequence conservation throughout eukaryotes, and could imply that the roles of SND1 and SND3 are performed by other proteins in metazoa. Notably, deletion of *SND3* in yeast led to impaired growth and generally more pronounced substrate mislocalisation compared to *SND1* and *SND2* knockouts[3]. In addition, the SND3 protein is an order of magnitude more abundant than SND1 or SND2[13]. Whilst it has been shown that yeast SND3 has an additional role in the formation of nucleus-vacuole junctions upon glucose starvation[14], these observations could also indicate that SND3 plays a more pivotal role in fungal membrane protein biogenesis than currently appreciated.

In this work, we describe a cryo-electron microscopy (cryo-EM) structure of a ribosome-associated SND3 translocon from the thermophilic fungus *Chaetomium thermophilum*. SND3 forms a complex with the biogenesis factors CCDC47, the SEC61 translocon and TRAPα beneath the ribosome tunnel exit. CCDC47 and SEC61β prevent nascent chain access to the SEC61 translocon, whilst SND3 has the hallmark characteristics of a membrane insertase. Structural and sequence comparisons indicate that the SND3 translocon is a multipass translocon (MPT) for co-translational insertion of IMPs in fungi, euglenozoan parasites and other eukaryotic lineages that lack a metazoan-like MPT.

## Results

### SND3 co-purifies with the PAT complex and ribosomes

In order to isolate native SND3-associated complexes, we generated a genetically modified *C. thermophilum* strain expressing SND3 with a C-terminal TwinStrep tag (*ct*SND3-TwinStrep). Notably, we observed that the PAT complex components, CCDC47 and Asterix, were co-eluted with detergent-solubilised *ct*SND3-TwinStrep after affinity purification and further size-exclusion chromatography (SEC) (Supplementary Fig. 1A). The PAT complex has recently been identified as an intramembrane chaperone complex and a core component of the ribosome-bound MPT, responsible for the biogenesis of multipass membrane proteins in metazoa[15–17]. Given that CCDC47 binds to ribosomes[16,18], we then looked whether ribosomes were co-purified with *ct*SND3. Indeed, we found that a small proportion of *ct*SND3 was pelleted with ribosomes after ultracentrifugation (Supplementary Fig. 1B), and a low-resolution map

from negative stain EM showed extra density at the ribosome tunnel exit (Supplementary Fig. 1C).

Given that a large population of *ct*SND3 was observed in the ribosome-free fraction (Supplementary Fig. 1B), we reasoned that the tag may not be accessible for affinity purification when *ct*SND3-TwinStrep is associated with ribosomes. Therefore, we further integrated CCDC47 with an N-terminal FLAG tag (*ct*CCDC47-FLAG) into our *C. thermophilum* strain. Indeed, a tandem affinity purification of *ct*SND3-TwinStrep then *ct*CCDC47-FLAG produced a stoichiometric complex between *ct*SND3 and the PAT complex, but with no ribosomes associated (Supplementary Fig. 1D). Consequently, we performed a single FLAG-immunoprecipitation (IP) of *ct*CCDC47-FLAG followed by ribosome pelleting, similar to a strategy previously used to isolate the MPT[17]. The results showed that *ct*SND3-TwinStrep was once again co-purified with *ct*CCDC47-FLAG (Fig. 1A and Supplementary Fig. 1E), but now both proteins were enriched in the ribosome-containing sample (Supplementary Fig. 1F). Mass spectrometry (MS) analysis of samples taken after either FLAG-IP or ribosome pelleting further revealed the presence of the SEC61 translocon components and translocon-associated protein alpha (TRAPα) in both samples (Fig. 1A). Whilst Asterix was present after the FLAG-IP of *ct*CCDC47-FLAG and also after the tandem purification with *ct*SND3-TwinStrep, it was notably absent after ribosome pelleting. Therefore, Asterix is either not present in ribosome-associated complexes or it dissociates during the pelleting. Nevertheless, the results show that *ct*SND3 associates with the complete PAT complex and that a complex comprising at least *ct*CCDC47 and *ct*SND3 interacts with ribosomes.

### Architecture of the ribosome-bound SND3 translocon from Chaetomium thermophilum

We next analysed the structure of ribosome-bound *ct*SND3/CCDC47 by single-particle cryo-EM, resulting in a final reconstruction at an overall resolution of 2.2 Å (Fig. 1B, Supplementary Fig. 2A, Table 1 and Supplementary Movie 1). In this map, the local resolution of the *C. thermophilum* 60S ribosomal subunit was around 2–3 Å, whilst the previously described rotation of the 40S ribosomal subunit[19] led to a lower resolution of 3–6 Å (Supplementary Fig. 3A). The map has density for a tRNA bound in the pe/E hybrid conformation as in the structure of the (TI)-POST-translational state[20] (Supplementary Fig. 3B), however there is no clear density for a nascent chain in the ribosome exit tunnel. Clearly resolved density within a detergent micelle positioned at the ribosome tunnel exit had a resolution of 3–6 Å and corresponded in part to the characterised structures of CCDC47 and the three subunits of the SEC61 translocon, SEC61α, SEC61β and SEC61γ (Supplementary Fig. 3C). The density features (Supplementary Fig. 4) enabled us to conduct model building and refinement starting with AlphaFold 3 (AF3)[21] models of intramembrane subcomplexes (Supplementary Fig. 3D–F). We firstly focused on the *ct*SEC61 translocon, which adopts its canonical position directly beneath the ribosome tunnel exit[22]. The conserved interactions between cytosolic loops within the C-terminal half of SEC61α[22] and uL23, eL19 and 26S rRNA within the 60S ribosomal subunit are well-resolved (Supplementary Fig. 5). As in previous structural studies[16,18], the cytosolic domain of *ct*CCDC47 is also well-resolved and interacts with both the ribosome and the *ct*SEC61 translocon, whereas its single TMD is poorly resolved, likely due to intrinsic flexibility (Fig. 1B and Supplementary Fig. 3C).

Alongside *ct*CCDC47 and the *ct*SEC61 translocon, two further densities for IMPs were present within the detergent micelle (Supplementary Fig. 3C). The first four-TMD density next to *ct*CCDC47 was confirmed to be *ct*SND3 after rigid body docking of an AF3 model for a *ct*CCDC47/SND3 complex (Supplementary Figs. 3E and 6A). *ct*SND3 and *ct*CCDC47 interact primarily via their cytosolic domains (Supplementary Fig. 6B), with electrostatic interactions stabilising the interface (Supplementary Fig. 6C). Like CCDC47 and the SEC61 translocon,

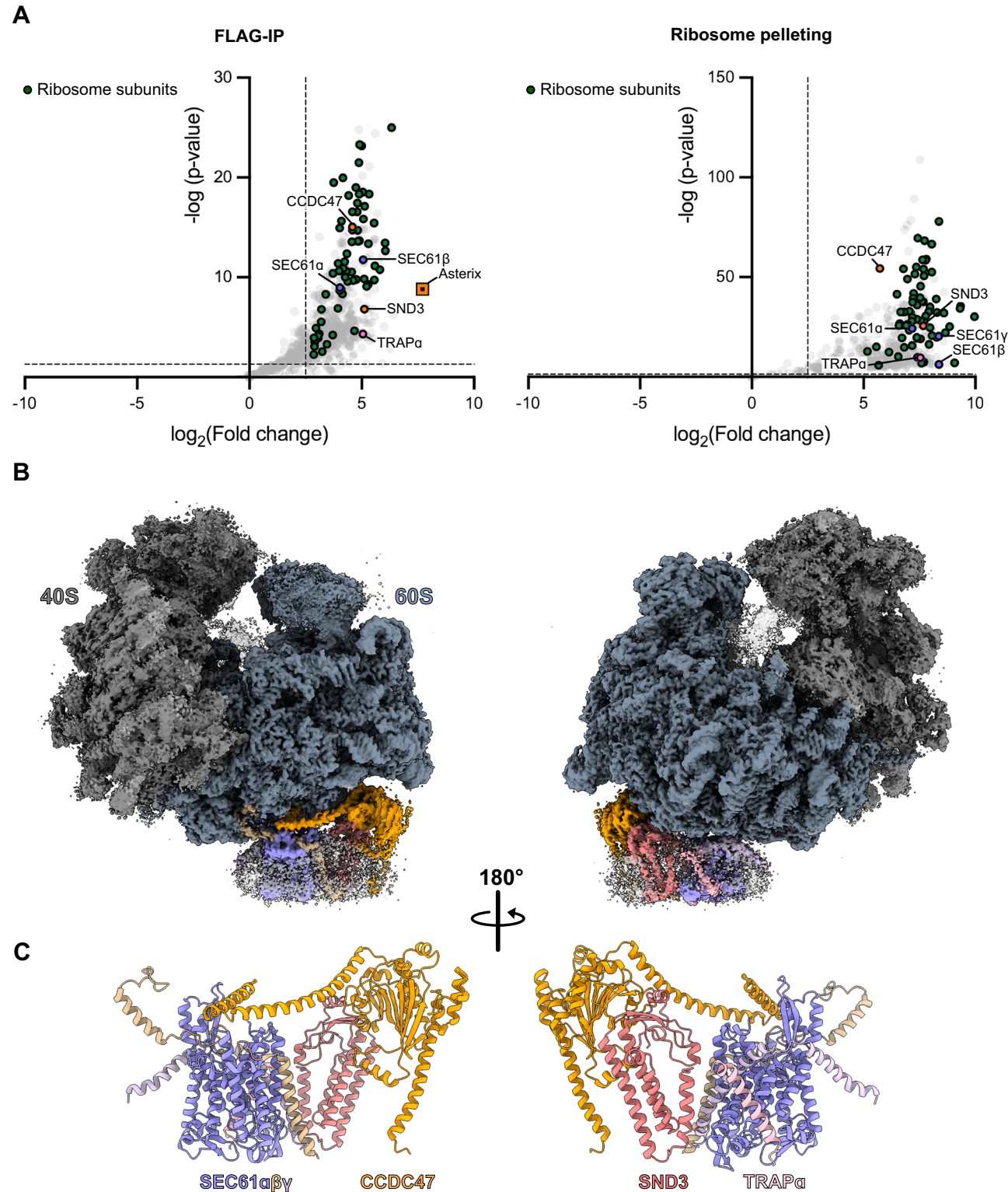

**Fig. 1 | Cryo-EM structure of the ribosome-bound SND3 translocon. A** Volcano plots showing proteins enriched in samples taken after FLAG-IP (left) and ribosome pelleting (right) from the *ct*CCDC47-FLAG/*ct*SND3-TwinStrep strain over equivalent control samples purified from wild-type *C. thermophilum*. Differential abundance analysis was conducted on four independent biological purifications. Components of the SND3 translocon/ribosome complex, which are significantly enriched (log₂(fold change) >2.5, *p*-value < 0.05) in our sample, are highlighted (pairwise comparison, see "Methods" section). Source data are provided in the Source Data file. **B** Cryo-EM density of the *C. thermophilum* ribosome-bound SND3 translocon complex. The ribosomal 40S and 60S subunits are coloured in grey and steel blue, respectively. The colouring of each component in the detergent micelle corresponds to the **C** model for the SND3 translocon, comprising the *ct*SEC61 translocon, *ct*CCDC47, *ct*SND3 and *ct*TRAPα.

## Table 1 | Cryo-EM data collection, refinement and validation statistics

| | Ribosome-bound SND3 translocon (EMD-52656; PDB 9I78) | Map with improved density for TRAPα luminal domain (EMD-52829) |
|---|---|---|
| **Data collection and processing** | | |
| Detector | Gatan K3 | |
| Magnification | 105,000× | |
| Voltage (keV) | 300 | |
| Calibrated pixel size (Å/pixel) | 0.837 | |
| Total dose (e⁻/Å²) | 60 | |
| Defocus range (μm) | −1.0 to −2.5 | |
| Symmetry imposed | C1 | C1 |
| Number of micrographs | 14,039 | |
| Initial number of particles | 2,108,591 | |
| Final number of particles | 119,533 | 31,235 |
| Map resolution (Å) | 2.2 | 2.6 |
| Resolution range | 2.2–4.3 | 2.3–7.1 |
| FSC threshold | 0.143 | 0.143 |
| **Model composition** | | |
| Non-hydrogen atoms | 133,843 | |
| Protein residues | 7583 | |
| Nucleotide | 3414 | |
| Ligands | | |
| Mg²⁺ | 379 | |
| SAC | 1 | |
| OMG | 1 | |
| **Refinement** | | |
| Initial model used (PDB ID) | 7OLC | |
| Map sharpening B-factor (Å²) | −52.4 | −52.0 |
| Model resolution (Å) FSC threshold (0.143) | 2.3 | |
| RMS deviations | | |
| Bond lengths (Å) | 0.004 | |
| Bond angles (°) | 0.604 | |
| MolProbity score | 1.59 | |
| Clash score | 8.47 | |
| Ramachandran plot | | |
| Favoured (%) | 97.33 | |
| Allowed (%) | 2.67 | |
| Outliers (%) | 0.00 | |
| Rotamer outliers (%) | 0.31 | |
| Average B-factors (Å²) | | |
| Protein | 102.41 | |
| Nucleotide | 112.70 | |
| Ligand | 97.52 | |

ctSND3 interacts with the 60S ribosomal subunit (Supplementary Fig. 6B), forming salt bridges with uL24 (Supplementary Fig. 6D). Additionally, a cluster of arginine and lysine residues in the ctSND3 cytosolic domain provides a highly positive surface for electrostatic interactions with the 26S rRNA (Supplementary Fig. 6E), further favouring the docking of the ribosome to the complex.

The second unassigned density near the ctSEC61 translocon harbours a single TMD and an ER luminal domain visible at low contour levels (Supplementary Figs. 3C and 6F). Further data processing with a focused mask on this area generated a cryo-EM map with enhanced density resembling metazoan TRAPα (Supplementary Fig. 2B) [23–26]. An AF3 model of the ctSEC61 translocon and ctTRAPα complex was subsequently generated and fit into the map, confirming this density represents ctTRAPα (Supplementary Figs. 3F and 6G). The interaction interface within this complex is formed exclusively by the luminal end of the ctTRAPα TMD and the hinge domain of ctSEC61α (Supplementary Fig. 6H).

The final structural model thus comprises a complex of SND3, CCDC47, TRAPα and the SEC61 translocon and is hereafter referred to as the SND3 translocon (Fig. 1C and Supplementary Movie 1). The canonical interaction between the SEC61 translocon and the ribosome is highly suggestive of a role for the SND3 translocon in co-translational protein translocation.

### CCDC47 and SEC61β prevent nascent chain access to the SEC61 channel

To gain further insights into the function of the SND3 translocon, we next conducted a detailed structural analysis of its components and subcomplexes, starting with the conserved complex formed by the SEC61 protein-conducting channel and CCDC47. As in the metazoan MPT [16,18], our structure shows that the C-terminal latch helices of ctCCDC47 are sandwiched between the ribosome tunnel exit and the entrance of the ctSEC61α channel (Fig. 2A). Similarly, the C-terminus of ctCCDC47 is positioned within the mouth of the ribosome exit tunnel (Fig. 2B), thereby narrowing this opening and altogether significantly obstructing access of the nascent chain to the cytosolic vestibule of ctSEC61α. Notably, ctSEC61α is present in a closed conformation (Supplementary Fig. 7). As observed previously [16], the ctCCDC47 latch helices closely interact with the N-terminal half of ctSEC61α at the cytosolic loop between TMD2 and TMD3 (Fig. 2C) and in this position would clash with the open state of the channel (Supplementary Fig. 7). In addition, our structure reveals a further tight interaction interface mediated by two salt bridges between the second ctCCDC47 latch helix and the C-terminal half of ctSEC61α within its cytosolic ribosome-binding domain (Fig. 2C). ctCCDC47, therefore, effectively braces both halves of ctSEC61α, which would actively prevent the opening of the channel through rotation of the N-terminal half away from the ribosome-bound C-terminal half. Therefore, overall, ctCCDC47 acts analogously to within the metazoan MPT to prevent co-translational protein translocation through the ctSEC61 translocon.

Strikingly, we found an additional density within our reconstruction that interacts with the ribosome and extends to the entrance of ctSEC61α (Supplementary Fig. 8A). Using automatic modelling by Model Angelo [27], we determined that this density corresponds to the cytosolic N-terminal region of ctSEC61β (Fig. 2A), the structure of which has not been resolved in other SEC61 translocon structures due to its intrinsic disorder. Our structure firstly reveals that residues 1–63 of ctSEC61β (hereafter called the ribosome-binding domain), which include an α-helix, are anchored to the ribosome (Supplementary Fig. 8B). Specifically, this interface involves salt bridges and hydrogen bonds with the ribosomal proteins eL19 and eL31 respectively, in addition to electrostatic interactions with 26S rRNA (Supplementary Fig. 8C). Therefore, SEC61β makes a significant unappreciated contribution to ribosome binding by the SEC61 translocon.

Surprisingly, we further discovered that residues 64–91 of ctSEC61β (hereafter called the vestibule binding domain) are sandwiched between the ctCCDC47 latch helices and the entrance of the ctSEC61α channel (Fig. 2A, D). The structural plasticity of this domain allows it to adopt a U-turn conformation within the cytosolic vestibule of ctSEC61α (Fig. 2E). ctSEC61β enters the vestibule above the lateral gate and makes extensive contacts with ctSEC61α, first with the

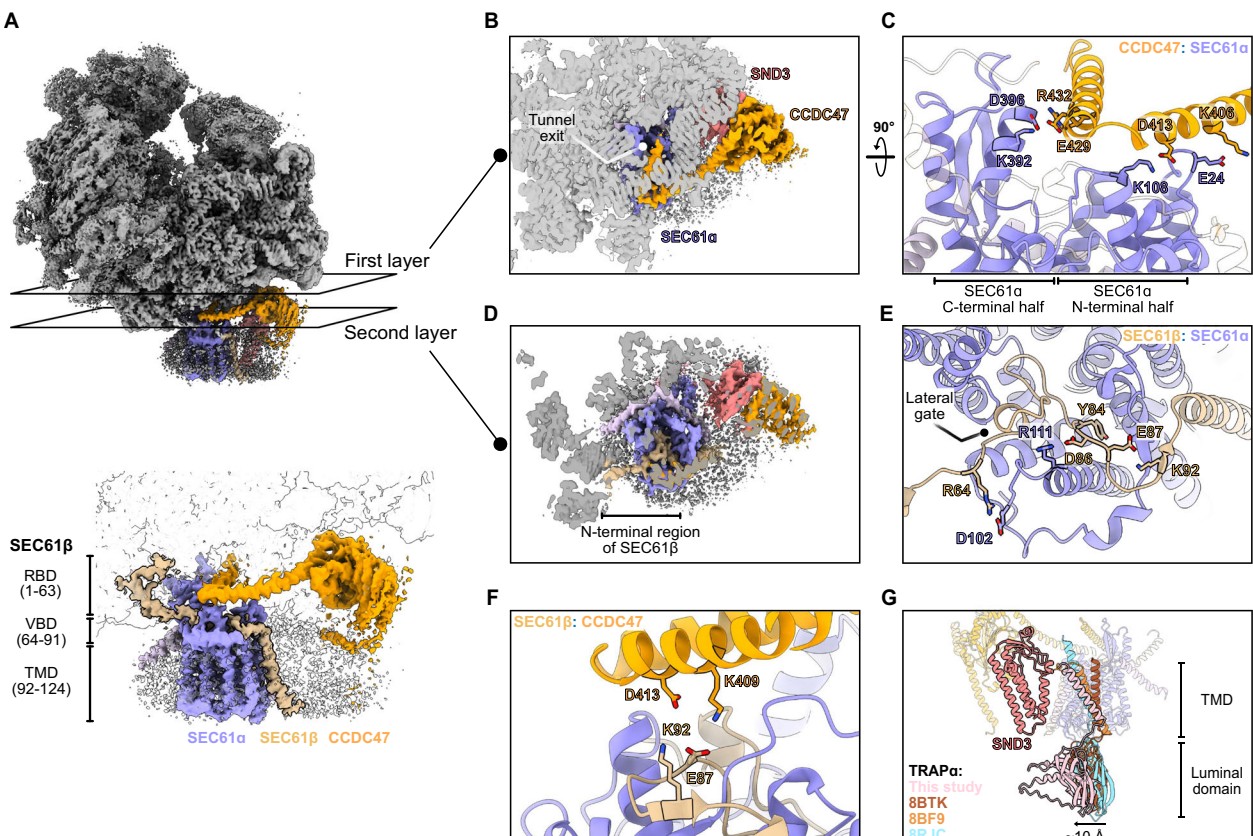

**Fig. 2 | The interaction partners of SEC61α. A** (Top) The positions of two slices through the cryo-EM map are indicated, corresponding to the inhibitory interactions between *ct*SEC61α and the latch helices of *ct*CCDC47 (first layer) or *ct*SEC61β N-terminus (second layer). (Bottom) Zoom-in of the same view indicating the ribosome binding domain (RBD), vestibule binding domain (VBD) and TMD of *ct*SEC61β. **B** The first layer slice from (**A**) viewed from above the cytosolic entrance of the *ct*SEC61α channel. **C** Detailed view of the electrostatic interactions between *ct*SEC61α and the latch helices of *ct*CCDC47. **D** The second layer slice from (**A**) viewed from above the cytosolic entrance of the *ct*SEC61α channel. **E** Detailed view of the interactions between *ct*SEC61α and the *ct*SEC61β vestibule binding domain. **F** Detailed view of the interface between the *ct*SEC61β vestibule binding domain and *ct*CCDC47. **G** Structural comparison of TRAPα between the SND3 translocon and other translocon complexes. *ct*SEC61α is superimposed with SEC61α from PDB 8BTK (RMSD 1.18 Å over 273 atoms), PDB 8BF9 (RMSD 0.95 Å over 151 atoms) and PDB 8JRC (RMSD 1.09 Å over 223 atoms). TRAPα homologues are opaque, and *ct*SND3 and *ct*TRAPα models from the SND3 translocon are outlined. The relative movement of the *ct*TRAPα luminal domain towards *ct*SND3 is indicated by an arrow.

C-terminal half and then the N-terminal half, before emerging at its TMD. Within the interface, *ct*SEC61β R64 and D86 form intermolecular salt bridges with SEC61α, Y84 inserts between SEC61α TMD3 and TMD4, whilst E87 and K92 form an intramolecular salt bridge to stabilise the SEC61β conformation (Fig. 2E). The *ct*SEC61β vestibule binding domain also approaches the latch helices of *ct*CCDC47, with potential electrostatic interactions mutually strengthening their binding within the complex (Fig. 2F). Therefore, *ct*SEC61β and *ct*CCDC47 appear to act synergistically to keep the SEC61 translocon out of action, with the *ct*SEC61β vestibule binding domain similarly blocking nascent chain access to the *ct*SEC61α channel and potentially providing a second brace to impede its opening.

## Structural insights into fungal TRAPα

Whilst the metazoan TRAP complex (a heterotetramer comprising α, β, γ and δ subunits) has an established role in co-translational protein translocation by the SEC61 translocon, it has only very recently been appreciated based on homology searches that fungi are also likely to have reduced-complexity TRAP comprising just the α subunit[25]. Our structure of the SND3 translocon now confirms that *ct*TRAPα is associated with the SEC61 translocon and adopts a similar arrangement to metazoan TRAPα[23–26]. Namely, *ct*TRAPα has an N-terminal domain positioned beneath the SEC61 channel and interacts with the *ct*SEC61α luminal hinge domain via its TMD (Supplementary Fig. 6H). As observed in some metazoan TRAPα structures[24,25], the TMD is notably tilted by ~45° with respect to the membrane plane, pointing its cytosolic end towards the tip of the 5.8S rRNA helix 7 and, uniquely to our structure, the N-terminus of SND3 (Fig. 2G). In metazoa, the C-terminus of TRAPα anchors the protein to the ribosome, even when the TMD is displaced from SEC61α[25,28]. Although *ct*TRAPα at least has the conserved residue W257 involved in ribosome binding (Supplementary Fig. 9), we observe no density in our map at its expected binding site[23,25], suggesting the interaction may be less robust in this system. Within the luminal domain, *ct*TRAPα also has some of the hydrophobic residues shown to be functionally important in metazoa (Supplementary Fig. 9)[23]. Interestingly, this domain is shifted by ~10 Å away from the *ct*SEC61α channel towards *ct*SND3 (Fig. 2G) and as such could be less poised to receive substrates from the closed channel. However, the luminal domain is poorly resolved and therefore likely to exhibit flexibility, perhaps due to a lack of stabilising interactions with a TRAPβ domain.

## SND3 has the features of a membrane insertase

We next analysed the features of the structurally uncharacterised component, SND3, within the complex. Our structure shows that *ct*SND3 has four TMDs, of which TMD1 and TMD4 are both shorter than is typical for IMPs (17 amino acids vs approximately 24 amino acids for a standard TMD[29]) and do not traverse completely to the cytosolic side of the membrane (Fig. 3A). In addition, TMD2, TMD3 and TMD4 are tilted relative to the membrane normal. In contrast to its short luminal loops, SND3 contains two long claw-shaped cytosolic

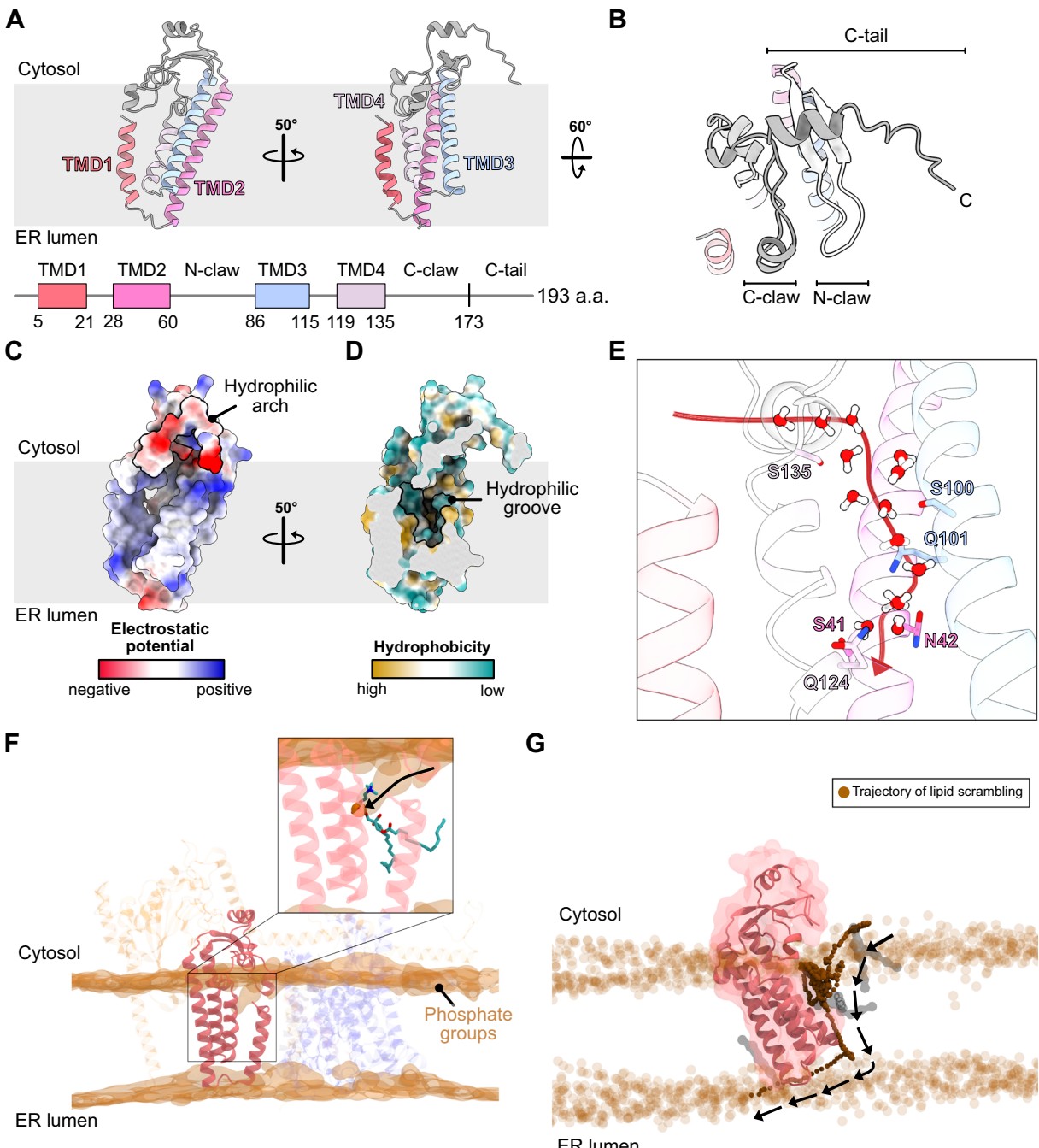

**Fig. 3 | SND3 has the features of a membrane insertase. A** (Top) Cartoon representation of the *ct*SND3 structure with the four TMDs coloured differently. (Bottom) Topology diagram showing the corresponding residue numbers for each TMD and the delineation between the C-claw and C-tail. **B** View of the *ct*SND3 cytosolic loops. **C** Electrostatic analysis of *ct*SND3 viewed as in (**A**) (left) performed in UCSF-ChimeraX[59] (minimum: −0.41 V, maximum: 0.30 V), highlighting the negatively charged hydrophilic arch formed by the cytosolic loops. **D** Hydrophobicity analysis of *ct*SND3 viewed as in (**A**) (right) performed in UCSF-ChimeraX (minimum: −28.41, maximum: 24.01), revealing a membrane-embedded hydrophilic groove formed by TMD2, TMD3 and TMD4. **E** Close-up of the *ct*SND3 hydrophilic groove showing hydrophilic residues and the distribution of water molecules in a snapshot of an equilibrated atomistic MD simulation. The red arrow indicates the potential direction of water movement from the cytosol into the hydrophilic groove. **F** Atomistic MD simulation of the SND3 translocon in a

membrane comprising 51% POPC, 36% POPE and 13% POPI. A snapshot of the SND3 translocon is shown (cartoon representation), where SND3 is opaque and other components are transparent. The density of phosphate atoms of lipid molecules in the bilayer is rendered as a transparent brown isosurface. (Inset) A representative phospholipid with the headgroup bound to the hydrophilic groove and the black arrow indicating its potential trajectory from an undistorted cytosolic membrane leaflet. **G** The lipid scrambling activity of *ct*SND3 identified in coarse-grained MD simulations. The atomistic SND3 structure (cartoon representation) is super-imposed on the coarse-grained representation (transparent surface). Opaque brown spheres represent the time-trace of the PO4 bead in the headgroup of a scrambled phospholipid (Supplementary Movie 2). Translucent brown spheres denote PO4 beads of the remaining phospholipids in the lipid bilayer. The tail group of the scrambling lipid is shown at three time points (grey surface) to represent the orientation before, during and after flipping.

loops that dip down towards the membrane plane, whereby the first claw (N-claw) is located between TMD2 and TMD3 and the second claw (C-claw) follows TMD4 (Fig. 3A, B). The cytosolic region after the C-claw (C-tail) comprises a short helix that caps the N-claw and an unstructured C-terminus that points towards the membrane plane. The conformation of these cytosolic elements is further stabilised by interactions with the *ct*CCDC47 cytosolic domain and the ribosome (Supplementary Fig. 6B). Analysis of the electrostatic potential showed that the N-claw and C-tail are highly negatively charged, together creating a hydrophilic arch on the cytosolic side of the membrane (Fig. 3C).

Interestingly, hydrophobicity analysis of *ct*SND3 further revealed a membrane-embedded hydrophilic groove within the cytosolic leaflet of the membrane formed by TMD2, TMD3 and TMD4 (Fig. 3D). Although no charged residues are present, these TMDs contribute many polar residues to the groove, including S41 and N42 of TMD2, S100 and Q101 of TMD3 and Q124 and S135 of TMD4 (Fig. 3E). Strikingly, such a hydrophilic groove formed by three TMDs is a hallmark feature of the Oxa1 superfamily of membrane insertases, where it provides the route for IMP substrate insertion[8]. However, the fold and topology of *ct*SND3 are notably different from those of the Oxa1 superfamily members, with the TMDs and extramembranous loops having a significantly different arrangement (Supplementary Fig. 10). Therefore, it appears that SND3 has the features to classify it as a distinct membrane insertase.

### SND3 disrupts the lipid bilayer and provides a pathway for lipid scrambling

We further noticed in our cryo-EM reconstruction that the detergent micelle around *ct*SND3 is 15 Å thinner than around other regions of the protein complex, particularly in the region of the short TMDs (Supplementary Fig. 11A). Therefore, we performed atomistic and coarse-grained molecular dynamics (MD) simulations of our SND3 translocon structure within a lipid bilayer. In general, the membrane-embedded protein domains and their interaction interfaces were stable over the course of the simulations, whilst the cytosolic domains of *ct*SEC61β and *ct*CCDC47 showed significant mobility, presumably due at least in part to the absence of their stabilising interactions with the ribosome (Supplementary Fig. S12). Notably, the N-claw and C-tail within the hydrophilic arch of *ct*SND3 exhibited some flexibility (Supplementary Fig. 13A), which correlates with their lower local resolution in the cryo-EM map (Supplementary Fig. 3C and Supplementary Movie 1). Furthermore, these cytosolic elements interact with the lipid headgroups via charged residues (e.g. E77, E78, R185, K189) (Supplementary Fig. 13B) and remain tethered to the surface of the membrane during the course of atomistic MD simulations (Supplementary Fig. 13C). In contrast, the *ct*SND3 C-claw, which comprises hydrophobic residues at its helical tip, penetrates the lipid bilayer and maintains a stable conformation (Supplementary Fig. 13A, B). Therefore, interactions between the *ct*SND3 cytosolic domains and the membrane appear to make an appreciable contribution to protein stability and conformation.

The atomistic MD simulations further revealed that the cytosolic leaflet of the membrane was dramatically distorted in the region adjacent to the short TMD1 and TMD4 of *ct*SND3, causing local membrane thinning (Fig. 3F). Specifically, phospholipid headgroups are drawn in to interact with the hydrophilic groove of *ct*SND3 within the membrane plane, with their hydrocarbon tails protruding at a distorted angle relative to the membrane normal (Fig. 3F inset and Supplementary Fig. 11B). In addition, the simulations showed that water molecules were able to interact with the polar residues of the *ct*SND3 hydrophilic groove, confirming it as a polar microenvironment in the membrane (Fig. 3E). Furthermore, a continuous network of water molecules connects to the hydrophilic groove through the thinned membrane region, highlighting a hydrophilic path from the

cytosol to the membrane core of *ct*SND3 (Fig. 3E and Supplementary Fig. 11C). Whilst water penetrates the luminal half of the *ct*SEC61α channel, it is excluded from the cytosolic vestibule, in accordance with the channel being closed in the structure (Supplementary Fig. 11C). Similarly, membrane perturbation around *ct*CCDC47 or the tilted TMD of *ct*TRAPα is negligible (Fig. 3F). Therefore, a hydrophilic moiety, such as a luminal loop of an IMP substrate, is most likely to enter the bilayer via the *ct*SND3 component of the SND3 translocon.

We further performed coarse-grained MD simulations to assess the trajectories of phospholipids over a longer timescale. Intriguingly, a number of phospholipids from the cytosolic leaflet were drawn into the membrane core such that the polar headgroup moved along the hydrophilic path indicated by the water network to sample the hydrophilic groove of *ct*SND3 (Fig. 3F inset and 3G; Supplementary Movie 2). Notably, this movement culminated in the flipping of some phospholipids between the two leaflets of the membrane (Fig. 3G and Supplementary Movie 2). Given that such lipid scrambling was recently found to be a general feature of protein insertases[30] and membrane thinning is a common strategy employed by the Oxa1 membrane insertases[31–34], our data altogether suggest that *ct*SND3 is a membrane insertase. Furthermore, the IMP substrate of a protein insertase, a hydrophobic TMD adjoined to a hydrophilic luminal loop, has amphipathic properties akin to a phospholipid. Therefore, we assert that the scrambled phospholipids observed in our MD simulations are effective substrate mimics, with the trajectory traced by the phospholipid headgroup highlighting a putative route for translocation of a short hydrophilic protein sequence by *ct*SND3.

### The SND3 translocon resembles a multipass translocon

Given that the interaction between the latch helices of *ct*CCDC47 and the closed *ct*SEC61 translocon is conserved with the metazoan MPT, we next compared our structure with those of the MPT via superimposition of the SEC61 translocon (Fig. 4A). This showed that the C-terminal latch helices of CCDC47 are positioned identically within the SND3 translocon and MPT[16], and that the N-terminal TMD and globular domain have a broadly similar location at the back of the translocon i.e. the opposite side to the lateral gate (Supplementary Fig. 14A, B). In addition, the *ct*CCDC47 TMD and globular domain together adopt the same structural fold in the SND3 translocon, aside from two small helical insertions in cytosolic loops (Supplementary Fig. 14C). Notably, however, these N-terminal domains of *ct*CCDC47 are significantly repositioned in the SND3 translocon, tilting by ~35° relative to CCDC47 in the MPT (Fig. 4B and Supplementary Fig. 14B). Strikingly, this change can be solely accounted for by differences in the helical connection between the globular domain and latch helices of CCDC47; in the SND3 translocon this connection is formed by one long α-helix that encompasses the first latch helix, whilst in the MPT it is formed by two distinct α-helices (one being the first latch helix) that adopt a ~130° angle relative to each other (Supplementary Fig. 14C).

One consequence of the repositioning of *ct*CCDC47 within the SND3 translocon is that the cytosolic region binds to a different surface of the ribosome, approaching the ribosomal protein eL32 and 26S rRNA H19 and H46 (Supplementary Fig. 14A), instead of eL6 and 28S rRNA H25 as in the MPT. A second consequence is that *ct*CCDC47 defines the boundary for a smaller lipid-filled cavity in the SND3 translocon and essentially occupies the same position as OPTI in the MPT (Fig. 4A). The major outcome is that *ct*SND3, a direct interaction partner of *ct*CCDC47, is pushed into the position of the Oxa1 superfamily insertase TMCO1 (Fig. 4C). Therefore, the proposed membrane insertases of the MPT and SND3 translocon, TMCO1 and *ct*SND3 respectively, occupy a similar site relative to the SEC61 translocon. Furthermore, within our superimposition, the resolved TMD of the inserting MPT substrate is located immediately adjacent to the hydrophilic groove of *ct*SND3 (Fig. 4D), suggesting a shared pathway for substrate insertion by the two translocon complexes.

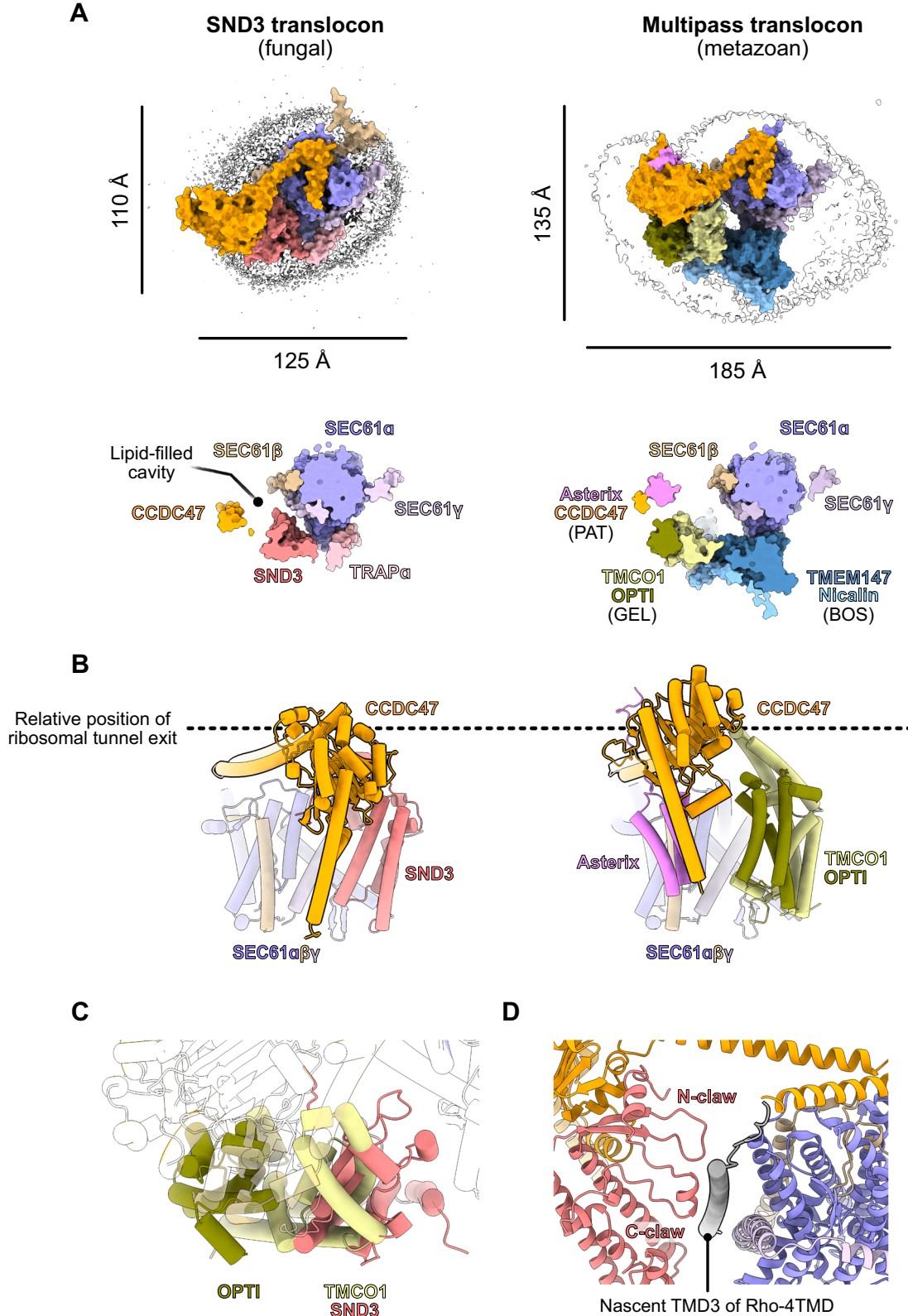

**Fig. 4 | The SND3 translocon structurally resembles the metazoan MPT.**
**A**–**D** Superimposition of models for the SND3 translocon and metazoan substrate-engaged MPT (PDB 7TUT) via SEC61α (RMSD 0.91 Å over 401 atoms). **A** (Top) The corresponding cryo-EM maps for the SND3 translocon and metazoan substrate-engaged MPT (EMD-26133) viewed from the cytosol. The dimensions of the detergent micelles are indicated, and conserved components are coloured the same. (Bottom) Surface representation of the models in the same view as the top panel, sliced at the level of the *ct*SND3 hydrophilic groove. **B** Cartoon representation of the models viewed along the membrane plane, illustrating tilting of *ct*CCDC47 in the SND3 translocon relative to the MPT. **C** View of the superimposed models showing positional overlap between *ct*SND3 from the SND3 translocon and the TMCO1 component of the GEL complex in the MPT. **D** The Rho-4TMD inserting nascent chain captured in the MPT structure sits adjacent to *ct*SND3 in the superimposition with the SND3 translocon.

It is unsurprising that the OPTI-TMCO1 (GEL) and TMEM147-Nicalin (BOS) subcomplexes of the MPT are 'missing' in the SND3 translocon (Fig. 4A), as their genes are absent from most fungi (Supplementary Fig. 15)[17]. Indeed, the position of the BOS complex is occupied in the SND3 translocon by TMD1 of *ct*SND3 and, interestingly, the TMD of *ct*TRAPα (Fig. 4A), in line with evidence that TRAPα and BOS compete for binding to the SEC61 translocon in metazoa[25,28]. Whilst Asterix is notably missing from the SND3 translocon structure, our biochemical and proteomics data demonstrate that *ct*Asterix does form a complex with *ct*CCDC47 and *ct*SND3 (Fig. 1A and Supplementary Fig. 1A). Superimposition of an AF3 model for the *ct*CCDC47/Asterix/SND3 ternary complex with the SND3 translocon suggests that *ct*Asterix would be accommodated, without steric clashes, on the periphery of the SND3 translocon, interacting with the *ct*CCDC47 TMD as in the metazoan PAT complex[16] (Supplementary Fig. 14D, E). Altogether, we therefore propose that the SND3 translocon represents a reduced-complexity fungal MPT, which conserves only the SEC61 translocon and PAT subcomplexes with the metazoan MPT. Crucially, structural adaptations in CCDC47 would enable the SND3 insertase to perform the same role as the metazoan GEL subcomplex in substrate insertion.

## Discussion

### Multipass IMPs as substrates for the SND3 translocon

Here, we reveal the cellular context of the SND3 component of the fungal SND pathway, showing it forms a complex with CCDC47, the SEC61 translocon and TRAPα beneath the ribosome tunnel exit. We propose that this SND3 translocon mediates co-translational protein insertion and find that it has an analogous organisation to the metazoan MPT, whereby the conserved components, CCDC47 and the SEC61 translocon, interact to prevent substrate insertion through SEC61α. In addition, SND3 has the characteristic features of a membrane insertase and occupies a similar position to the putative TMCO1 insertase within the metazoan MPT. Therefore, this work indicates that fungi have a designated MPT in the form of the SND3 translocon and implies that the preferred substrates for SND3 are multipass IMPs. Indeed, multipass IMPs are enriched with the purified SND3 translocon relative to single-pass IMPs (Supplementary Fig. 16A), although it cannot be ruled out that these represent direct interactors of the complex rather than substrates. Similarly, in a recent analysis of the yeast interactome, SND3 was predominantly found to cluster with multipass IMPs[35] (Supplementary Fig. 16B). In addition, deletion of *SEC61β*, which in our structure plays an important role in re-routing substrates to SND3, predominantly impedes biogenesis of multipass IMPs in yeast[36] (Supplementary Fig. 16C). Importantly, there is compelling evidence that human SND2 is involved in multipass IMP biogenesis[6,11,12], suggesting the SND pathway is aligned on this process.

Overall, the assignment of multipass IMP substrates to the SND pathway implies that the position of the first TMD relative to the N-terminus is likely to vary and to some extent justifies the broad substrate specificity defined previously on these terms[3]. This would also corroborate the involvement of SRP as a targeting factor for co-translational insertion[6]. However, certain proposed substrates, such as GPI-anchored proteins or IMPs with C-terminal TMDs, are harder to reconcile, as they require post-translational insertion or appreciable translocation through the SEC61 translocon. As elucidation of these substrates was based on cellular depletion of SND components via siRNAs or gene knockouts[3,9,11], their biogenesis could have been affected by compensatory or indirect effects. To this end, it is interesting that *SEC61β* knockout effects the biogenesis of several multipass IMPs involved in GPI-anchored protein biosynthesis and thus could indirectly impair this process[36] (Supplementary Fig. 16C). In addition to the co-translational insertion by the SND3 translocon proposed here, it also cannot be ruled out that SND3 can interact with the post-translational form of the SEC61 translocon to assist insertion of a different subset of substrates, as previously suggested[3].

### Evolution of multipass IMP biogenesis

These data further provide a missing puzzle piece in the evolution of multipass IMP biogenesis. Whilst the PAT complex and the SEC61 translocon are strictly conserved throughout eukaryotes, the mutually exclusive distribution of SND3 and TMCO1 between different taxa is striking; SND3 is predominantly found in fungi, whilst TMCO1, its interaction partner OPTI and the BOS complex are largely found in metazoa and plants (Fig. 5A and Supplementary Fig. 15). This suggests that TMEM109, which in terms of sequence and structure does not resemble fungal SND3, is unlikely to be the previously reported human SND3 homologue[11]. Interestingly, the non-overlapping division of SND3 and TMCO1 homologues between taxa also extends to early-branching eukaryotes (Fig. 5A and Supplementary Fig. 15). Specifically, stramenopiles and euglenozoa, including the parasitic *Trypanosoma* and *Leishmania* spp., have SND3, whilst alveolata have TMCO1. Unusually, amoebozoa like *Dictyostelium discoideum*, whose evolution branched just before fungi and metazoa, appear to have both insertases. It is even more intriguing that SND3 has a distinct fold for a membrane insertase, yet the TMCO1 membrane insertase belongs to the phylogenetically ancient Oxa1 superfamily found also in bacteria and archaea. Therefore, it altogether seems likely that SND3 was acquired by an early eukaryote and TMCO1 evolved from an archaeal Oxa1 predecessor[37,38], allowing two different forms of the MPT to develop in distinct lineages. Notably, the fungal MPT has a smaller lipid-filled cavity and is markedly less complex in terms of the number of components than the metazoan MPT. As there are fewer fungal multipass IMPs (yeast: 785, human: 2483) with, on average, fewer TMDs (yeast: median 6, human: median 7)[39], it is perhaps not surprising that fungi require a smaller, less sophisticated MPT. Nevertheless, defining the selective pressures that have dictated the evolution of either a fungal or metazoan-like MPT by different eukaryotes now presents a compelling research question.

In the context of the SND pathway, it is interesting that only SND2 is conserved between fungi and metazoa, suggesting that the recently assigned function of human TMEM208 in accelerating the release of substrates from SRP is also likely to be applicable to yeast SND2[6]. Whilst in current models, SRP hands over substrate to either the EMC or SEC61 translocon[40,41], the putative interaction between SND2 and SND3[3] could allow direct recruitment of or handover to the MPT after SND2-triggered release of multipass IMPs.

### The fungal MPT reveals additional accessory factors

Our data provide a clear view of how fungal TRAPα binds to the SEC61 translocon and, unlike in the metazoan MPT, is compatible with association of the other MPT components, likely due to the absence of steric clashes with a BOS complex[25,28]. The assigned functions of metazoan TRAP are largely linked to co-translational insertion of particular substrates via the SEC61 translocon, namely signal peptide insertion[42,43], interaction with translocated substrates[44], glycosylation by oligosaccharyltransferase A (OSTA)[45] or assisting membrane protein topogenesis[46]. Therefore, the interplay of TRAPα with a closed SEC61 translocon is unclear; however, the positioning of the luminal domain closer to SND3 in our structure makes it tempting to speculate that it could also interact with substrates inserted via this route. Furthermore, it is interesting that the single TRAPα subunit in fungi fulfils the role of a heterotetramer in metazoa or a heterodimer in plants, providing a simplified system to probe the mechanistic details of TRAP function in future work.

We also gain a major advance in understanding the role of SEC61β within the translocon complex, revealing that it binds the ribosome and acts together with CCDC47 to hinder nascent chain

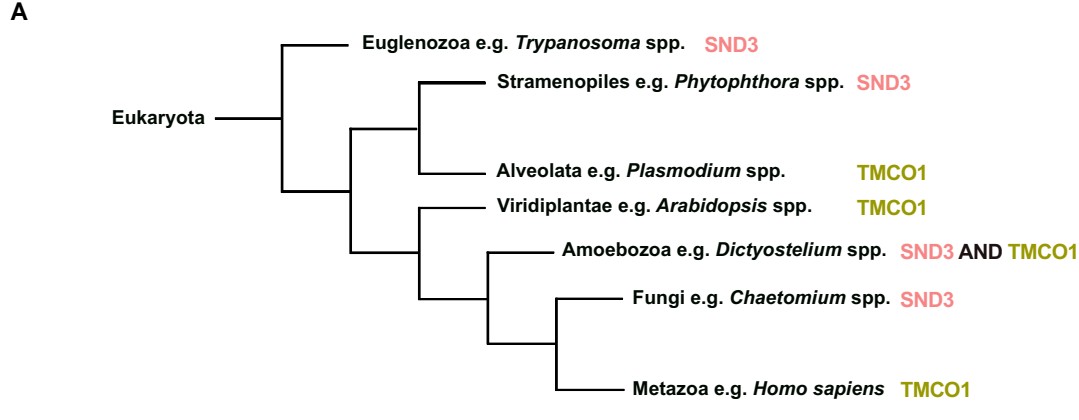

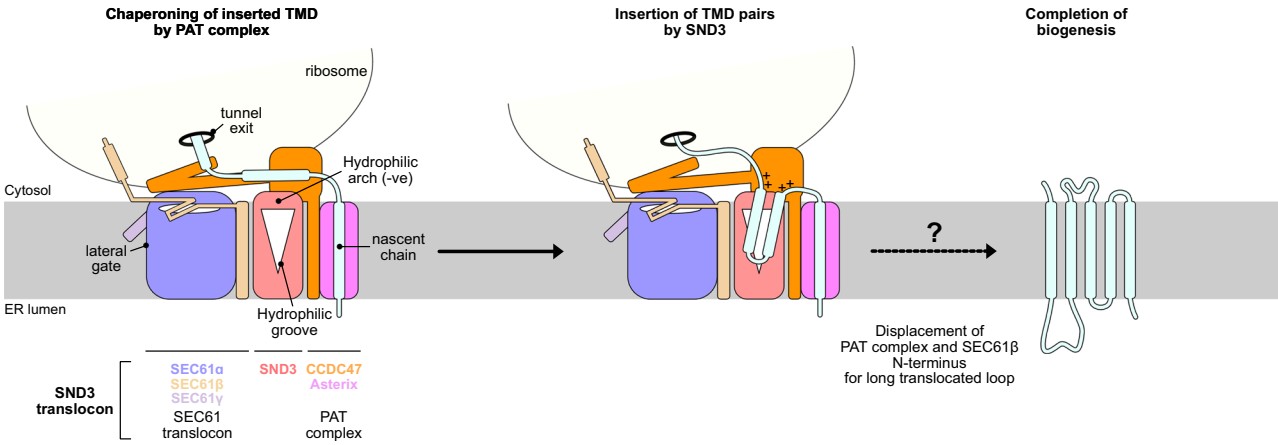

**Fig. 5 | Multipass IMP biogenesis by the SND3 translocon. A** Phylogenetic distribution of SND3 and TMCO1 insertases in selected eukaryotes. The predominant insertase(s) in each taxon were identified via searches for SND3 and TMCO1 homologues in OrthoDB v12.0[90], as illustrated in Supplementary Fig. 15. **B** Proposed mechanism of multipass IMP biogenesis by the SND3 translocon.

access to the SEC61α channel. Given that residues involved in these interactions are largely conserved in eukaryotes (Supplementary Fig. 17A), it is likely that this SEC61β arrangement is not unique to *C. thermophilum*; as similar density is not observed in the metazoan MPT structures[16,18], it is possible the N-terminus was resolved in this instance either due to the absence of nascent chain within the fungal MPT or to the overall improved resolution observed for this thermostable MPT, as has previously been observed for structures of other complexes[47,48]. Indeed, AF3 models for both the human and yeast SEC61 translocons predict an analogous interaction between the cytosolic vestibule of SEC61α and conserved residues within the N-terminus of SEC61β (Supplementary Fig. 17B). Furthermore, interactions between the N-terminus and both ribosomes[49] and CCDC47[18] have previously been observed for mammalian homologues, suggesting a conserved functionality for these domains in higher eukaryotes. Interestingly, despite differences in sequence and topology to eukaryotic SEC61β, the cytosolic loop of the *Thermus thermophilus* functional homologue SecG was previously seen to occlude the SecY channel[50], indicating bacteria could also employ a similar mechanism to block the translocon. Given that SEC61β is a constitutive part of the SEC61 translocon and the observed binding to SEC61α is unlikely to be compatible with translocation through the channel, further work is required to determine whether the N-terminus interacts with other states of the translocon and how this is regulated. As SEC61β is required for the biogenesis of a subset of, often multipass, IMPs[36] and the N-terminus has been shown to cross-link to early emerging nascent chains[51,52], engagement with SEC61α may occur in response to particular substrates. Of note, the reported phosphorylation of SEC61β[53,54], which has been shown to affect biogenesis of some yeast substrates[36], could also serve to regulate the predominantly electrostatic interactions of the ribosome and vestibule binding domains observed in our structure.

## Mechanistic implications for co-translational insertion by the SND3 translocon

A notable finding of this study is that SND3 is likely to be a membrane insertase with an atypical fold, harbouring a characteristic membrane-embedded hydrophilic groove that destabilises the membrane and scrambles phospholipids. By highlighting a route for the translocation of a short hydrophilic protein sequence across the ER membrane, these data altogether suggest that SND3 inserts TMDs by a mechanism highly analogous to that proposed for the Oxa1 family members[8]. Specifically, the luminal loop of a nascent IMP substrate would first be drawn from the cytosol along a hydrophilic path into the hydrophilic groove of SND3. This loop would eventually traverse the relatively short hydrophobic distance of the thinned bilayer to the ER lumen, accompanied by the energetically favourable partitioning of the hydrophobic TMD into the membrane. The negatively charged hydrophilic arch of SND3 may retain positively charged cytosolic loops of substrate IMPs and, via the positive inside rule[55], promote correct substrate topology.

The SEC61 translocon is closed in our captured state of the fungal MPT, suggesting co-translational insertion of multipass IMPs proceeds

via SND3. This suggests a mechanism analogous to that proposed for the metazoan MPT (Fig. 5B)[16], whereby SND3 translocates the luminal loop between two adjacent TMDs of an IMP, allowing them to insert as pairs into the lipid-filled cavity. Furthermore, the conserved PAT complex could be expected to have a similar role in chaperoning early inserted TMDs while they await association with later-inserted TMDs[15]. As for the insertases of the Oxa1 family[8], the hydrophilic groove of SND3 is likely to only translocate short luminal loops (<50 amino acids). Therefore, insertion of longer loops would require concomitant dissociation of CCDC47 and SEC61β and opening of the SEC61α channel for translocation. In contrast to the metazoan multipass translocon being assembled in response to its specific substrates[16,17], we have captured the SND3 translocon in the absence of a nascent chain. Whilst this could be attributed to the thermostability of this *C. thermophilum* complex, the SND3 translocon is present here in an idle state, and requires biochemical and structural characterisation in the presence of a translocating nascent chain in order to further dissect its mechanism of substrate insertion. Nevertheless, we made a significant advance in aligning research on the SND pathway and the MPT, setting a paradigm for understanding the evolution and mechanisms of multipass IMP biogenesis.

## Methods

### Generation of C. thermophilum genomically modified strains

For the generation and integration of native affinity-tagged constructs into the *C. thermophilum* genome, the genes encoding *ct*SND3 (CTHT_007540) and *ct*CCDC47 (CTHT_0024800) were amplified from *C. thermophilum* genomic DNA together with their 1 kb or 343 bp upstream promoter regions, respectively, using the primers in Supplementary Table 1. The *ct*SND3 fragment with a C-terminal TwinStrep tag and the *ct*CCDC47 fragment with an internal FLAG tag after residue T60 were respectively subcloned into the pNK51 vector (with the *ERG1* thermostable selection marker) or the pNK130 vector[56] (with the *HygB* thermostable selection marker), both kind gifts from N. Kellner and E. Hurt (Heidelberg University). The resulting plasmids were linearised by restriction enzyme digest, concentrated through ethanol precipitation and stored at −20 °C until further use.

*C. thermophilum* strains were transformed with the linearised plasmids as described previously in detail[56,57]. Briefly, *C. thermophilum* wild-type spores (DSM1495) were re-germinated on a CCM agar plate at 52 °C for 2–3 days. Mycelia were scraped from the plate, and small pieces were used to inoculate CCM media for scaling up to a 150 mL culture for transformation. Mycelia were harvested, and the cell wall was digested using an enzyme blend comprising pectinases, beta-glucanase, protease and arabinanase (Vinotaste Pro, Novozymes). The resulting protoplasts were filtered, washed and subsequently used for PEG-mediated transformation with 10 µg of the linearised *ct*SND3-TwinStrep plasmid DNA. The transformed protoplasts were plated on a double-layer CCM-Sorbitol (CCMS) agar plate, comprising a top layer of CCMS agar and a bottom layer of CCMS agar supplemented with 0.5 µg/mL terbinafine, and grown at 50 °C for 2–3 days. Colonies were then re-plated on a CCM agar plate supplemented with 0.5 µg/mL terbinafine to confirm stable gene integration. The linearised *ct*CCDC47-FLAG plasmid DNA was integrated into the *ct*SND3-TwinStrep strain as described above, except spores were initially re-germinated on a CCM agar plate supplemented with 0.5 µg/mL terbinafine and transformed protoplasts were recovered on CCMS plates supplemented with 200 µg/mL hygromycin B in the bottom layer at 42 °C for 6–7 days. Colonies were re-plated on a CCM agar plate supplemented with 75 µg/mL hygromycin B. Stable transformants of both the *ct*SND3-TwinStrep and *ct*SND3-TwinStrep/*ct*CCDC47-FLAG strains were used to inoculate small-scale cultures, and expression of the epitope-tagged target proteins was confirmed by western blotting with specific antibodies, as described below.

### Western blotting

5 µL sample was run on a 4–20% gradient Tris-Glycine SDS-polyacrylamide (PA) gel (Novex) and then transferred to a 0.2 µm PVDF membrane within a Power Blotter Select Transfer Stack (Invitrogen). The membrane was blocked in 2% (w/v) milk powder (Roth) dissolved in TBST (20 mM Tris-HCl (pH 8.0), 150 mM NaCl, 0.1% (v/v) Tween-20 (Merck)) at 4 °C overnight. For detecting *ct*SND3-TwinStrep, the blocked membrane was incubated with murine monoclonal anti-Strep antibody (IBA, 2-1507-001) at a ratio of 1:2500 at 4 °C for 1 h. For *ct*CCDC47-FLAG, the membrane was incubated with murine monoclonal anti-FLAG M2 antibody (Sigma–Aldrich, F3165-1MG) at a ratio of 1:1000. The blotted membranes were washed 3× with TBST at 4 °C for 10 min and then incubated with Peroxidase AffiniPure Goat Anti-Mouse IgG (H + L) (Jackson ImmunoResearch, 115-035-146) at a ratio of 1:5000 at 4 °C for 1 h. After washing 3× in TBST at 4 °C for 10 min, the membranes were developed using Western Lightning Ultra Chemiluminescent Substrate (Perkin Elmer) and visualised using the ChemiDoc Imaging System (BioRad).

### Membrane preparation from C. thermophilum

*C. thermophilum* strains were cultivated in 4 L CCM media and grown in a rotary incubator at 52 °C with agitation at 85 rpm for 18 h. Mycelia were harvested, washed and dried, then immediately frozen in liquid nitrogen and stored at −70 °C until further use. Mycelia were resuspended in NB-250S buffer (20 mM HEPES (pH 8.0), 100 mM potassium acetate, 10 mM magnesium acetate, 250 mM sucrose), supplemented with 10 µg/mL cycloheximide (CHX), 100 U RNase inhibitor RiboLock (Thermo Scientific) and cOmplete EDTA-free protease inhibitor cocktail (Roche). The mycelia were lysed by bead beating using the Retsch MM400 with 0.5 mm glass beads and a frequency of 30 Hz for 4 × 5 min. The lysate was cleared by centrifugation at 1000 × *g* at 4 °C for 10 min, followed by centrifugation at 10,000 × *g* at 4 °C for 10 min to remove residual cell debris, nuclei and mitochondria. The supernatant was subjected to ultracentrifugation at 185,500 × *g* at 4 °C for 90 min, and the resulting membrane pellet was resuspended in the NB-250S buffer before flash-freezing in liquid nitrogen and storage at −70 °C until further use.

### Initial purification and characterisation of ctSND3-associated complexes

Membranes prepared from the *C. thermophilum* *ct*SND3-TwinStrep strain were solubilised in NB-250S buffer, 10 µg/mL CHX, 100 U RNase inhibitor RiboLock, cOmplete EDTA-free protease inhibitor cocktail, and 0.5% (w/v) LMNG (Anatrace) at 4 °C for 1 h. Solubilised membranes were centrifuged at 10,000 × *g* at 4 °C for 20 min, and the supernatant was mixed with 250 µL StrepTactin resin (IBA) at 4 °C overnight. The resin was washed 5× with four-bed volumes of NB-250S buffer and 0.01% (w/v) LMNG, then eluted with two rounds of 1 h incubation at 4 °C with four-bed volumes of NB-250S buffer, 0.01% (w/v) LMNG and 2.5 mM *d*-Desthiobiotin (Sigma–Aldrich). The two elution fractions were pooled, concentrated using a Vivaspin 20 centrifugal concentrator 100 kDa MWCO (Cytiva) and subjected to SEC with a Superdex 200 increase 3.2/300 (Cytiva), equilibrated with SEC buffer (20 mM HEPES (pH 8.0), 100 mM potassium acetate, 10 mM magnesium acetate and 0.01% (w/v) LMNG). SEC fractions were separated on 4–20% gradient Tris-Glycine SDS-PA gels (Novex), then visualised with Coomassie blue staining and western blotting. Coomassie-stained gel bands were identified by MS.

To purify potential *ct*SND3-associated ribosomes, the eluate from the StrepTactin resin was layered on a sucrose cushion (20 mM HEPES (pH 8.0), 100 mM potassium acetate, 10 mM magnesium acetate, 600 mM sucrose and 0.01% (w/v) LMNG) at a volume-to-volume ratio of 1:3 and centrifuged at 150,000 × *g* at 4 °C for 16 h. The pellets were resuspended in 20 mM HEPES (pH 8.0), 100 mM potassium acetate,

10 mM magnesium acetate and 0.01% (w/v) LMNG. The sample concentration was estimated by the absorbance at 260 nm ($A_{260}$; 1 unit ~60 µg/mL) and used at 90 µg/mL for negative stain EM analysis.

### Negative stain EM analysis

Support grids (Cu 400 mesh, Sigma–Aldrich G5026-1VL) were coated with a layer of 3 nm continuous carbon using a Leica EM ACE 600 and glow-discharged using a Pelco EasiGlow device. 4 µL sample was applied to the grid and stained with 0.2% (w/v) uranyl formate. The grids were then manually blotted and air-dried overnight. 216 micrographs were collected on a FEI Tecnai Spirit electron microscope at 120 keV using Digital Micrograph software (Gatan). A nominal magnification of 30,000×, corresponding to a calibrated pixel size of 2.1 Å, was used. Using cryoSPARC v4.4.0[58], 187,541 particles were picked using the blob picker, then extracted in a box size of 320 pixels and Fourier-cropped to 160 pixels. 12,417 particles were selected from 2D classification and subjected to ab initio 3D reconstruction. The resulting map was visualised using UCSF-ChimeraX[59].

### Tandem purification of the ctSND3/ctCCDC47 complex for MS

Membranes prepared from the *C. thermophilum* *ct*SND3-TwinStrep/*ct*CCDC47-FLAG strain were solubilised and purified with StrepTactin resin as for the *ct*SND3-TwinStrep strain. The two elution fractions were pooled and incubated with 50 µL anti-FLAG M2 affinity gel (Millipore) at 4 °C overnight. After two washes with 20 bed volumes of NB-250S buffer and 0.01% (w/v) LMNG, the complex was eluted with two rounds of 1 h incubation with three bed volumes of NB-250S buffer, 0.01% (w/v) LMNG and 500 µg/mL 3×FLAG peptide (Sigma–Aldrich) at 4 °C. The eluate was separated on 4–20% gradient Tris-Glycine SDS-PA gels (Novex), visualised with Coomassie blue staining before MS identification of stained gel bands.

### Purification of the C. thermophilum SND3 translocon for cryo-EM

Membranes from the *C. thermophilum* *ct*SND3-TwinStrep/CCDC47-FLAG strain were solubilised in NB-250S buffer, 10 µg/mL CHX, 100 U RNase inhibitor RiboLock, cOmplete EDTA-free protease inhibitor cocktail and 1.5% (w/v) digitonin (PanReac AppliChem) at 4 °C for 1 h. Solubilised membranes were centrifuged at $10,000 \times g$ at 4 °C for 20 min, and the supernatant was mixed with 100 µL anti-FLAG M2 affinity gel (Millipore) at 4 °C overnight. The gel was then washed 3× with 10 bed volumes of NB-250S buffer and 0.25% (w/v) digitonin, and the complex was eluted with three rounds of 1 h incubation with two bed volumes of NB-250S buffer, 0.25% (w/v) digitonin and 500 µg/mL 3×FLAG peptide (Sigma–Aldrich) at 4 °C. For ribosome pelleting, the eluate was layered on a 600 µL sucrose cushion (20 mM HEPES (pH 8.0), 100 mM potassium acetate, 10 mM magnesium acetate, 600 mM sucrose and 0.25% (w/v) digitonin) and centrifuged at $250,000 \times g$ at 4 °C for 2 h. The pellets were resuspended in 20 mM HEPES (pH 8.0), 100 mM potassium acetate, 10 mM magnesium acetate and 0.25% (w/v) digitonin. The sample concentration was estimated by the $A_{260}$ and used at 3 mg/mL for cryo-EM grid preparation. Samples from throughout the purification were separated on 4–20% gradient Tris-Glycine SDS-PA gels (Novex), then visualised with Coomassie blue staining and western blotting.

### Purification of the C. thermophilum SND3 translocon for MS

Samples of the *C. thermophilum* SND3 translocon were purified as outlined for cryo-EM with the following changes. A control purification from wild-type *C. thermophilum* membranes was always performed in parallel, and four independent purifications were obtained for both control and *ct*SND3-TwinStrep/CCDC47-FLAG containing membranes. All buffers were prepared in DEPC-treated water, and membranes were treated with 10,000 units of micrococcal nuclease in the presence of 1 mM calcium acetate at 25 °C for 10 min before solubilisation. This

reaction was terminated by adding 2 mM EGTA and 20 U/mL super-aseIN (Invitrogen). From this point forward, all buffers contained 10 µg/mL CHX and 5 U/mL superaseIN. Samples of the FLAG-IP eluate and ribosome pellet were stored at −70 °C before further processing.

### Cryo-EM grid preparation and data collection

Holey carbon support grids (Quantifoil 300 mesh, Cu R1.2/1.3) were coated with a layer of 3 nm continuous carbon using a Leica EM ACE 600 and glow-discharged using a Pelco EasiGlow device. 3 µL purified sample was applied to the grid and adsorbed for 15 s, before blotting for 4 s at 4 °C and 100% humidity (blot force: 0) and plunge-freezing in liquid ethane using a Vitrobot Mark IV (Thermo Scientific). Data acquisition was performed on a Titan Krios G3i transmission electron microscope (Thermo Scientific) operated at 300 keV in EFTEM mode and equipped with a BioQuantum-K3 imaging filter (Gatan). 14039 movies were collected in counting mode using EPU 3.8 software (Thermo Scientific) with an energy filter slit width of 20 eV, a defocus range of −1.0 to −2.5 µm at a nominal magnification of 105,000×, corresponding to a calibrated pixel size of 0.837 Å. The dose rate was adjusted to ~15 e⁻/pix/s and a total dose of 60 e⁻/Å² was distributed over 60 fractions with a total exposure time of 3.0 s.

### Cryo-EM data processing

All raw movies were subject to motion correction with the Relion 4.0[60] implementation of MotionCor2[61], applying dose-weighting and $5 \times 5$ patch-based alignment. Corrected micrographs were then transferred to cryoSPARC v4.4.0[58] for patch-based contrast transfer function (CTF) estimation. The micrographs whose CTF_fit_to_Res parameters were between 2.5 and 4 Å were selected for particle picking with the template-free blob picker. The picked particles were extracted with a box size of 640 pixels and Fourier-cropped to 160 pixels for iterative rounds of 2D classification.

Approximately 0.8 million particles were selected as input for a 5-class ab initio reconstruction with C1 symmetry, followed by heterogeneous refinement to generate five distinct classes. Particles from four classes were used for a non-uniform refinement to generate an initial model. A mask focusing on the detergent micelle was generated in UCSF-ChimeraX[59] and then used for 3D variability analysis (3DVA) in cryoSPARC v4.4.0 to generate five clusters of structural ensembles. Two of the five clusters, which together accounted for ~50% of the total particles, yielded maps with clear density for the ribosome, SEC61 translocon and CCDC47. We therefore pooled the particles from these two classes for a further round of focused 3D classification without alignment (class = 10). 119,533 particles corresponding to three of the ten clusters were re-extracted with the same box size and Fourier-cropped to 512 pixels, then subjected to non-uniform refinement with CTF refinement to yield the final cryo-EM map with a nominal resolution of 2.20 Å (Supplementary Fig. 2A). Local resolution analysis was performed using ResMap[62].

To improve the resolution of the luminal domain of TRAPα, particles from cluster 4 in the first round of 3DVA were selected for an additional round of 3DVA, with a focused mask on the back region of SEC61 (class = 5). After assessing the quality of the resulting maps, 31,235 particles from cluster 5 were extracted with a box size of 640 pixels and Fourier-cropped to 512 pixels, and then subjected to non-uniform refinement to generate a 2.64 Å cryo-EM map (Supplementary Fig. 2B).

### Model building and refinement

The published atomic model of the *C. thermophilum* ribosome (PDB 7OLC) was used to derive the starting model for the 60S subunit. Models for *C. thermophilum* CCDC47/SND3 and SEC61 translocon/TRAPα complexes were generated by AlphaFold 3.0[21] and initially placed into the cryo-EM map using UCSF-ChimeraX 1.7.1[59]. Precise positions for individual domains were then obtained by rigid body docking of the SEC61α/TRAPα subcomplex, the SEC61β TMD, SEC61γ,

SND3 and residues 76–370, 371–417 and 417–447 of CCDC47 using UCSF-ChimeraX or Coot 0.9.8[63]. Notably, more continuous density for the CCDC47 TMD and SND3 C-terminus was observed in the unsharpened final reconstruction. The initial model of the N-terminal region of *C. thermophilum* SEC61β was built by Model Angelo in Relion 5.0[27]. Density features confirmed that the protein sequence after residue 97 of SEC61β (VLVLSLVFIFSVVALHVIAKITRKFSS) differs from UniProt entry G0SCD9 and, as such, is interpreted as a database error additional to the *C. thermophilum* ribosomal proteins previously reported[20]. The fitted models were further refined by a combination of real-space refinement in Phenix 1.21[64] and manual refinements in Coot or ISOLDE[65] in UCSF-ChimeraX to obtain a convergent model. The final model was assessed by MolProbity[66]. Structural visualisations and superimpositions were performed in UCSF-ChimeraX.

## MS sample preparation
Gel bands (two sets of 5 samples each, total: 10 samples) were processed according to the protocol for the Tryptic Digestion Kit (Thermo Scientific, 89871) using ABC buffer (FluKa, 40876-50G), TCEP (Sigma–Aldrich, 646547-10X1ML), iodoacetamide (Thermo Scientific, A39271) and Trypsin (Promega, V5280). Proteolytic peptides were stored at −20 °C until liquid chromatography (LC)-MS analyses.

Pulldowns were conducted in four independent biological experiments (total: 16 samples). Solution SND3 translocon samples were digested using S-TRAP micro cartridges (Protifi) according to the manufacturer's instructions (reduction using TCEP and alkylation using iodoacetamide as above). Peptides were further desalted and purified on Isolute C18 SPE cartridges (Biotage) and dried in a concentrator (Eppendorf). Proteolytic peptides were stored at −20 °C until LC-MS analyses.

## LC-MS acquisition
*Orbitrap Eclipse*. After solubilisation in 0.1% formic acid (FA) in acetonitrile (ACN)/water (95/5, v/v), peptides were loaded onto an Acclaim PepMap C18 capillary trapping column (particle size 3 μm, L = 20 mm) and separated on a ReproSil C18-PepSep analytical column (particle size = 1.9 μm, ID = 75 μm, L = 25 cm, Bruker) using a nano-HPLC (Dionex U3000 RSLCnano) at a temperature of 55 °C. Trapping was carried out for 6 min with a flow rate of 6 μL/min using a loading buffer composed of 0.05% trifluoroacetic acid in $H_2O$. Peptides were separated by a gradient of water (buffer A: 0.1% FA in water) and ACN (buffer B: 80% ACN, 20% water, 0.1% FA) with a constant flow rate of 400 nL/min. The gradient went from 4% to 48% buffer B in 45 min. All solvents were LC-MS grade and purchased from Riedel-de Häen/Honeywell. Eluting peptides were analysed in data-dependent acquisition mode on an Orbitrap Eclipse mass spectrometer (Thermo Scientific) coupled to the nano-HPLC by a Nano Flex ESI source. MS1 survey scans were acquired over a scan range of 350–1400 mass-to-charge ratio (m/z) in the Orbitrap detector (resolution = 120k, automatic gain control (AGC) = 2e5, and maximum injection time: 50 ms). Sequence information was acquired by a ddMS2 OT HCD MS2 method with a fixed cycle time of 2 s for MS/MS scans. MS2 scans were generated from the most abundant precursors with a minimum intensity of 5e3 and charge states from two to five. Selected precursors were isolated in the quadrupole using a 1.4 Da window and fragmented using higher-energy collisional dissociation (HCD) at 30% normalised collision energy. Orbitrap MS2 data were acquired using a resolution of 30k, an AGC of 5e4 and a maximum injection time of 54 ms. Dynamic exclusion was set to 30 s with a mass tolerance of 10 parts per million (ppm).

*TimsTOF HT*. After solubilisation in 0.1% FA in ACN/water (95/5, v/v), samples were subjected to LC-MS/MS analysis on a nanoElute2 (Bruker) system, equipped with a C18 analytical column (15 cm * 150 μm, particle size: 1.5 μm, Bruker) coupled to a timsTOF HT mass spectrometer (Bruker). Samples were loaded directly onto the analytical column with twice the sample pick-up volume with buffer A.

Peptides were separated on the analytical column at 60 °C with a flow rate of 500 nl/min with the following gradient: 2–38% B in 21.0 min, 38–95% B in 0.5 min and constant 90% B for 3.5 min with buffer A (0.1% FA in water) and buffer B (0.1% FA in ACN).

For gel band analyses, peptides eluting from the column were ionised online using a captive spray ion-source and analysed in DDA-PASEF mode with a cycle time of 1.1 s with ten PASEF-MS/MS events. Spectra were acquired over the mass range from 100 to 1700 m/z and a mobility window from 0.6 to 1.6 Vs/cm².

For S-TRAP digests, peptides eluting from the column were ionised using a captive spray ion-source and analysed in DIA-PASEF mode. Precursors in a range of m/z 475–1000 and a 1/K0 range of 0.85–1.27 were scanned in 21 scans with an isolation window of 25 Th per step, with no overlap to neighbouring windows and a total cycle time of around 0.97 s.

## MS data processing
DDA data analysis (gel bands) was performed in FragPipe v21.1 using MSFragger 4.0 for database searches[67]. DIA data analysis (solution SND3 translocon samples) was performed using DIA-NN (ver. 1.9.2) for database searches[68]. Raw files were recalibrated, search parameters automatically optimised and searched against the Uniprot proteome for *C. thermophilum DSM1495* (obtained 2023-12-08).

All database searches were restricted to tryptic peptides with a length of 7–30 amino acids, up to two missed cleavages and with a minimum of one unique peptide per protein group. Carbamidomethylation of Cysteine was set as a fixed modification, and oxidation of Methionine, as well as N-terminal acetylation, were set as variable modifications. Percolator was used to estimate the number of false positive identifications, and the results were filtered for a false discovery rate (FDR) < 0.01.

Differential abundance analyses on DIA data of in-solution samples were conducted using the MS-DAP script (v1.2.1), with pairwise comparisons between control and *ct*SND3-TwinStrep/CCDC47-FLAG samples using MSqRob with qvalue thresholds of 0.01 and a minimum of 1 unique peptide detected in at least 2 of the 4 independent biological replicates per condition.

## All-atom simulations
**Molecular modelling.** For the MD simulations, we used the atomic coordinates of the SND3 translocon as determined by cryo-EM. Our simulation model included the SEC61α, SEC61β, SEC61γ, CCDC47, SND3, and TRAPα chains. We used the *membrane builder*[69,70] module of the CHARMM-GUI[71] server to construct three independent initial simulation models. In each model, the proteins were oriented with respect to the lipid bilayer membrane using the PPM 2.0 server[72]. The plane of the membrane was kept perpendicular to the z-axis. The composition of the bilayer was taken according to the ER membrane of *Neurospora crassa*[73] (50.51% POPC, 36.34% POPE, 13.15% POPI), where the two leaflets are symmetric (similar lipid composition). $Na^+$ and $Cl^-$ ions were added to establish a salt concentration of -150 mM in an overall neutral system. The resulting system components are listed in Supplementary Table 2. The rectangular simulation box had dimensions of $20.6 \times 20.6 \times 13.4$ nm³. The proteins, lipids and ions were modelled using the CHARMM36m force field[74]. For water, we used the CHARMM-modified TIP3P model[75].

**Atomistic MD simulations.** MD simulations were performed using the GROMACS-2022.4 simulation package[76]. Each system was first energy minimised using the steepest descent algorithm with harmonic restraints on protein backbones (force constant $k = 4000$ kJ mol$^{-1}$ nm$^{-2}$), side-chains ($k = 2000$ kJ mol$^{-1}$ nm$^{-2}$), lipids ($k = 1000$ kJ mol$^{-1}$ nm$^{-2}$), and dihedrals ($k = 1000$ kJ mol$^{-1}$ rad$^{-2}$). Energy minimisation was followed by a systematic sequence of equilibration runs (following the protocol recommended by CHARMM-GUI[71]) before

the final production run. The details of each step are given in Supplementary Table 3. This process ensured a thorough equilibration of the system. The final production simulation was performed for 300 ns (time step dt = 2 fs) under NpT conditions with a data output interval of 1 ps. The equations of motion were integrated using the leap-frog algorithm. The temperature and pressure were maintained at 300 K and 1 bar, respectively, using the stochastic velocity rescaling (v-rescale) thermostat[77] and stochastic exponential relaxation (c-rescale) barostat[78]. In the NpT simulations (both equilibration and production), a semi-isotropic pressure coupling was applied with a time constant ($\tau_p$) of 5 ps and compressibility of $4.5 \times 10^{-5}$ bar$^{-1}$. Temperature coupling was applied separately on the protein, lipid and solvent (water and ions) atoms with time constants ($\tau_t$) of 1 ps each. Protein bonds involving hydrogen atoms and internal degrees of freedom of water molecules were constrained using the LINCS[79] and SETTLE[80] algorithms, respectively. Short-range Coulomb and van der Waals interactions were computed up to a distance cut-off of 1.2 nm. Long-range electrostatic interactions were treated with the Particle Mesh Ewald method[81] with a grid spacing of 0.12 nm.

### Coarse-grained simulations

**Molecular modelling.** The atomistic model of the translocon complex was first converted to the MARTINI3[82] coarse-grained model using the *martinize2*[83] script. Elastic bonds with force constants of 500 kJ mol$^{-1}$ nm$^{-2}$ were applied between the protein atoms within a distance range of 0.5 and 0.9 nm in individual chains to maintain their secondary structures during the simulations. The protein complex was embedded in a lipid bilayer and solvated using the *insane*[84] script. Lipid composition was taken from the all-atom simulations[73]. The resulting system consisted of a total of 1442 POPC, 1037 POPE, and 375 POPI molecules. 1194 sodium and 1218 chloride beads were added to neutralise the system and maintain a salt concentration of 150 mM. The system contained 109531 water beads (1 martini CG water bead effectively represents 4 water molecules). The box volume was $30 \times 30 \times 20$ nm$^3$.

**Coarse-grained MD simulations.** The coarse-grained MARTINI3 simulations were performed using the GROMACS-2022.4 package[76]. Following an energy minimisation using the steepest descent algorithm, the system was equilibrated in two steps, each for 1 μs. In the first step, harmonic restraints were applied on the protein and lipid beads (k = 1000 kJ mol$^{-1}$ nm$^{-2}$). The position restraints were removed in the second step. Both the equilibration steps were performed under NpT conditions with semi-isotropic pressure coupling of 1 bar with $\tau_p$ = 4 ps using the Berendsen barostat[85]. Temperature was maintained at 300 K using the v-rescale thermostat[77] ($\tau_p$ = 1 ps) with separate temperature coupling on protein-membrane and water-ion groups. The electrostatic and van der Waals interactions were computed using the MARTINI recommended settings[86]. The final production run was performed for 10 μs (analysed in 10 blocks of 1 μs each to estimate statistical uncertainty), with a data output interval of 200 ps. In this run, the Parinello–Rahman barostat[87] was used for pressure coupling, with $\tau_p$ = 12 ps. All simulations were performed using the leap-from algorithm with a dt of 20 fs.

All MD simulation data were analysed and visualised using the codes listed in Supplementary Table 4, and an MD simulation checklist for data reproducibility is provided as Supplementary Table 5.

### Reporting summary

Further information on research design is available in the Nature Portfolio Reporting Summary linked to this article.

## Data availability

Coordinates for the *Chaetomium thermophilum* SND3 translocon bound to the 60S ribosomal subunit have been deposited in the Protein Data Bank under accession number 9I78. The respective cryo-EM map has been deposited in the Electron Microscopy Data Bank under accession code EMD-52656, and the cryo-EM map with improved density for the TRAPα luminal domain under accession code EMD-52829. The MS proteomics data have been deposited to the ProteomeXchange Consortium via the PRIDE partner repository[88] with the dataset identifier PXD060914. The MD simulation files are available on Zenodo as entry 16745187[89]. Source data are provided with this paper.

## Code availability

The analysis code for the MD simulation data is available on Zenodo as entry 16745187[89].

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

## Acknowledgements

We thank the Central Electron Microscopy Facility at the Max Planck Institute of Biophysics for providing cryo-EM infrastructure and technical support, particularly Susann Kaltwasser and Sonja Welsch. We also thank the Membrane Proteomics and Mass Spectrometry Facility at the Max Planck Institute of Biophysics for conducting MS experiments, particularly Imke Wüllenweber for support with sample preparation. We are grateful to Irmi Sinning and Werner Kühlbrandt for providing infrastructure and a network for discussion, and to Silke Adrian and Alena Zindel for support with culturing *C. thermophilum*. We acknowledge Nikola Kellner and Ed Hurt for the pNK51 and pNK130 plasmids. We thank Juan Castillo for support with computing and Birgit Wolf for media preparation. We also thank Sanjoy Paul for helping in setting up the coarse-grained MD simulations. M.A.M. was funded by the Max Planck Society and the Deutsche Forschungsgemeinschaft (DFG; German Research Foundation) through SFB 1507/P16 (project-ID 450648163). G.H. acknowledges DFG support through SFB 1507/P12. This project has received funding from the European Union's Horizon 2020 research and innovation programme under the Marie Skłodowska-Curie grant agreement No 101107937 (Acronym: StructureSND) to T-J.Y. Views and opinions expressed are, however, those of the author(s) only and do not necessarily reflect those of the European Union or European Research Executive Agency (REA). The authors thank the Max Planck Society for support.

## Author contributions

T-J.Y. and M.A.M. designed the study. T-J.Y. created *C. thermophilum* strains, purified protein samples, collected EM data, processed EM data and built the structural models. T-J.Y. and M.A.M. interpreted the structural data. T-J.Y., M.A.M. and J.D.L. analysed MS data. S.M. and G.H. performed and analysed MD simulations. T-J.Y. and M.A.M. wrote the manuscript. All authors contributed to the final version of the manuscript.

## Funding

## Competing interests

The authors declare no competing interests.
