## [Transparent Peer Review file · Nature Communications]

SND3 is the membrane insertase within a distinct SEC61 translocon complex

Corresponding Author: Dr Melanie McDowell

Version 1:

Reviewer comments:

Reviewer #1

(Remarks to the Author)

This is significant work and moves the field forward, as it reveals not only a previously undocumented fungal ER translocon, but provides many insights into the architecture, likely function and evolution of co-translational multipass translocons that are responsible for integrating multi-TMD containing proteins into the lipid bilayer during their biogenesis.

An integrated biochemical, single particle cryogenic-EM and Molecular Dynamics approach is used to analyze a ribosome-associated translocon complex isolated from the thermostable fungi, *Chaetomium thermophilum*. The authors have investigated the partners of SND3 by using a series of pull downs from genetically tagged strains including SND3-twin strep and then with a FLAG tagged partner, CCDC47, which copurifies with SND3 and is a known member of metazoan multipass translocons. They then purified ribosome associated SND3-CCDC47 in sufficient quantities and purity for single particle analysis, to obtain near atomic maps (avg. 2.2 and 2.6 Å resolution) for the large subunit and somewhat lower for the small subunit, which was not analyzed further as this was not the focus of this paper. Typically, regions of ribosome associated components are often imaged at a resolution which gets worse with distance from the primary binding site and this appears to be the case in this specimen. However, the local resolution is good enough for molecular docking of AlphaFold derived models of Sec61 subunits, CCDC47 and SND3 which provides a reasonable (conservative and accurate) description for the major areas of interest highlighted in the text and Figures (but see comments below). The ribosome complexes appear to be in a POST-like conformation with a tRNA in the pe/E-site (although occupancy is hard to assess, and no nascent chain (NC) density was observed in the Lsu tunnel.

Many important points arise from this analysis that are addressed clearly and appropriately in this manuscript. Briefly, a probable multipass translocon in fungi is comprised of membrane proteins, Sec61 (alpha,beta,gamma), SND3, CCDC47 and TRAP alpha. The fungal translocon has many structural similarities to a metazoan multipass translocon that also contains Sec61 and CCDC47, along with TMCO1 an insertase with some structural similarities to SND3, although the latter has a different fold and thus, SND3 represents a different family of insertases that are distinct from the canonical Oxa1 derived lineage. Hence, the likely role of SND3 has been identified in the context of a ribosome complex. In both translocons, CCDC47 has a domain inserted into the gap between Sec61 and the Lsu that would block nascent chain access to the canonical channel. Intriguingly, the Sec61 beta N-terminus is well resolved in the fungal complex and has both a ribosome binding component and also extends into the Sec61 vestibule wherein it may also block NC access. The presence of TRAP alpha (the lone subunit from the vertebrate 4 subunit complex that is present in fungi and presumably contains the active site in all organisms within this complex) is present in a similar but somewhat altered radial position on Sec61 that may position it to work in concert with other components in the multipass translocon.

The arrangement of transmembrane regions in the SND3 multipass translocon creates a lipid filled cavity that is smaller than in the metazoan complex, which could be involved in the integration of membrane protein TMDs, coordinated with SND3's putative role in being able to accommodate charged/polar regions given its charge distribution within a groove that penetrates partway across the membrane. The possible effects of SND3 on water structure and neighboring phospholipids (via their charged head groups) was evaluated with Molecular Dynamics and are consistent with the authors suggestion that local and partial bilayer disruption/disorder induced by the topology of SND3 may be critical to insertase activity, in the context of a nascent chain with multiple TMs during co-translational translocation.

Given the new structural and biophysical data, which is correlated with other published work, a reasonable and general mechanism is proposed for the fungal MPT that also brings to bear Asterix which was not present in the isolated structure but was shown to interact with SND3-TwinStrep in pulldown experiments; modeling indicates that Asterix would not be sterically hindered in binding to CCDC47, which suggest that it may be a more dynamic component. This work provides a unifying principal for the function of both fungal and metazoan MPTs.

A few technical comments to be addressed in this work:

With regards to map quality:

1. Table 1 : The much larger and better defined ribosome dominates resolution and validation statistics. It would be useful to have a section with local resolution range and validation statistics for ribosome associated components that are important to the focal point of the paper (Sec61 channel, CCDC47 and SND3). Perhaps a Q-factor, or similar metric for model fit into density (FSC0.5 model vs segmented maps) for the Alphafold3 model components and some description of changes in the models when they were "morphed" by refinement into the density (RMSDs).
2. The local resolution map figures are small and it's hard to judge the overall quality, ie where side chains are clearly resolved for the translocon components. From the shown Figures it seems like side chains would be resolved at the important interfaces described in the text, but a few examples of map quality in these regions would be useful to the reader, along with some description of the overall distribution of side chain densities and the regions where one is able to discern adjacent helical turns (cf. for the docking of TRAP alphas TMD etc. for example).
3. There was controversy in the early literature concerning whether the ribosome made a tight seal to the Sec61-like channel, while the first cryogenic 3D maps showed a sizable gap and suggested that the gap would allow access to this region that might facilitate co-translational folding of cytoplasmic domains of membrane proteins or some other functions. Clearly in this work the gap is used by accessory factors to modulate channel activity and promote integration of multipass membrane proteins that bypass the central pore of the Sec-like channel, while Sec61 alpha/gamma, still function as a ribosome docking site and organizational centerpiece to assemble a larger translocon with enhanced functions. Chaetomium Sec61 alpha-lsu interactions via the 6/7 and 8/9 loops play an important role in forming this translocon and in general, these interactions have not been resolved at fairly high resolution; it would be useful to present some data on this critical interface, which occurs in close proximity and within the tunnel exit.
4. pg 14, line 351 : "sequentially" is not correct, I think they mean "sequence-wise".

Reviewer #2

(Remarks to the Author)

Here, Yang and co-workers present a break-through structural study of the SND system, probably the least understood ER machinery for membrane protein targeting and insertion in eukaryotic cells. The manuscript starts with a procedure for isolation of the complex (together with the ribosome) followed by structure determination by cryo-EM and molecular dynamics simulations. The membrane-embedded SND component, Snd3, appears in association with the Sec61 translocon and CCDC47 protein, previously described as a part of membrane protein insertase. CCDC47 serves to stabilize the closed conformation of the Sec61 translocon, here being assisted by the cytoplasmic extension of the Sec61-beta subunit. The structural data is employed as an input for MD simulations, which confirm the distortion of the lipid bilayer and high lipid mobility at the interface of Sec61 and Snd3, which indirectly supports the hypothesis that Snd3 is the insertase for membrane proteins, that utilizes similar mechanism as Oxa1/YidC-type insertases.

The results of the study are of great interest, especially in light of the earlier publications on membrane protein insertases from mammalian cells by the groups of RS Hegde and R Keenan. The experimental data are solid and provide a unique, pioneering view on the SND pathway. My major comments are related to the data interpretation and discussion.

1. Differently to Sec61-BOS-GEL-PAT complex trapped in presence of a defined substrate protein, the Sec61-SND3 translocon is purified from the crude ribosome extract. As a result, no nascent protein is seen - neither in the membrane, not at the interface or even within the ribosome exit tunnel. That may be a result of averaging, but can also be interpreted as the insertase complex being in its idle state. The studies on the mammalian system showed that the multipass translocon is assembled in response to the specific nascent protein, so one should consider such scenario when evaluating the Snd3-Sec61 data. In particular, the Discussion on pages 15-16 sounds rather speculative, as the resolved structure is interpreted solely as an active state. Also the title/abstract should be adjusted, as the presented data (shown in Suppl. Figure 13) are not sufficient to claim the Snd3-Sec61 as a "multipass translocon".
2. In connection to the point above, I would recommend to address the visualized complex as Sec61-SND3 translocon/insertase or alike, as both elements are playing distinct roles, and an interplay upon insertion of substrates, e.g. switching between "back-of-Sec61" and Sec61-mediated "lateral gate" routes cannot be excluded.
3. Comparison of Snd3 and TMCO1 within the insertase complex does not look convincing, and may be toned down in the discussion (page 14). The proteins occupy quite different positions/ orientations relatively to Sec61, and also the role of TMCO1 within metazoan insertase complex is not clear yet.
4. The architecture of the Snd3 claw domains is quite unusual (clearly different to the helical extension of TMCO1/Oxa1 insertases), and so their position at the membrane interface is. Suppl. Figure 10 suggests certain flexibility within the "claws", but possibly more detailed explanation of their dynamics may be useful, to learn whether they stay associated with the

membrane or detach/displace towards the cytoplasm.

5. The transmembrane rod-shaped density appearing between Sec61-gamma and Sec61-alpha TMH 8, and so in proximity to the lateral gate, seems to be ignored in the description of the cryo-EM data.

6. For presentation of the AlphaFold3 prediction plots, e.g. Suppl. Figure 3D, the areas/regions for individual proteins/subunits should be indicated for clarity.

Reviewer #3

(Remarks to the Author)

In their paper SND3 is the membrane insertase within a fungal multipass translocon Yang et al. describe a new function of SND3 as a membrane protein insertase in the thus far ill-characterized SND pathway. This is based on a cryo-EM structure of a SND3:Sec61:CCDC47:TRAP complex together with MD simulations. Based on these data, the authors suggest the observed complex to correspond to the multipass translocon in higher eukaryotes.

Taken together, this study provides compelling evidence for SND3 being a TMD insertase in the ER membrane of fungi and reveals structural details of a supercomplex it engages in. Also, evolutionary insights into membrane protein biogenesis in different species can be derived from this study. A few points should be addressed before publication:

- A recent study has proposed that SND2 acts as a SRP release factor, a very different function as it seems the authors of this study now propose. This should be discussed in more detail. As of now it remains unclear, how "the recently assigned function of human TMEM208 in accelerating release of substrates from SRP is also likely to be applicable to yeast SND2"
- What exactly do the authors mean by "it is likely that this SEC61 β arrangement is not unique to *C. thermophilum* and has perhaps not been observed in previous structures due to differences in experimental setup" – which differences are they referring to, as this is indeed an unexpected observation.
- Although claimed, the authors do not formally show SND3 to be an insertase. Any functional data on this would significantly strengthen the impact of this work, but may be beyond the scope of the current study.
- The depiction of the MS data in Fig. 1A/B is unusual, normally an enrichment over control and the entire volcano plot should be shown.
- In Fig. 3 C, what is the designation "gate" based on? Is it mobile? Where different conformations observed?
- The model in Fig. 5 proposes pairwise TMD insertion; what evidence is this based on?

Reviewer #4

(Remarks to the Author)

This interesting manuscript presents a cryo-EM structure of the membrane insertase SND3 together with ribosome, Sec61, Trap-alpha and CCDC47. To my knowledge, it is the first experimental structure of SND3. The structure also reveals the previously unknown conformation of the N-terminal tail of Sec61beta suggesting that it establishes contacts with the ribosome. Sec61 is observed in a closed state. In the observed conformation, access to the Sec61alpha pore is blocked by CCDC47 and Sec61beta. Trap-alpha is sideways displaced toward SND3 relative to previous cryoEM structures (without SND3).

Also, atomistic and coarse-grained MD simulations reveal that the lipid bilayer is thinned next to SND3 and suggest that SND3 acts as a flippase.

The manuscript is very well written and illustrated, cites relevant previous work, and ties in nicely with the current literature. Several omissions are mentioned below.

Points to be addressed in a revision:

(1) Analysis of the MD simulations (which were understandably conducted without ribosome) is presented not in enough detail. Fig. S10 shows RMSF plots of all subunits. Fig. 3F shows membrane thinning. Movie #2 illustrates mobility of a phospholipid.

(a) I suggest that the authors also comment on the stability of the individual protein subunits (RMSD wrt. to starting structure) and on the stability of the contacts between subunits. The atomistic simulations are likely too short (300 ns) to be affected by the absence of stabilizing ribosome. However, the coarse-grained simulations could show deviations.

(b) p.26 line 721 suggests that coarse-grained simulations were carried out without position restraints in the second part of the CG simulations. How about restraints to stabilize the protein secondary structure elements?

(2) p.7 line 152: Why is ctSec61alpha "also" present in a closed conformation. Do you also see it in open conformations elsewhere? Or is another subunit also closed?

(3) p.9 line 204 "suggesting it could play a similar role in co-translational protein translocation". Do you suggest here that TRAPalpha affects translocation via SND3? This is later picked up in line 383 in the discussion section - which is the appropriate place for speculation.

(4) p.11 line 262: "our data provide strong evidence for ctSND3 being a membrane insertase". I don't agree to this. You do not present functional membrane-insertion assays. You connect your structural data to experimental work presented elsewhere. The MD simulations monitor the mobility of a phospholipid. This may resemble a typical SND3 cargo in some

aspects, but a phospholipid is certainly not a multipass integral membrane protein. This sentence should be reworded.

(5) p.13 line 342 "co-translational insertion ... shown here". Same thing. You do not present data from insertion assays. Hence, you cannot state this here.

(6) p.16 line 410 "these data altogether show that SND3 inserts TMDs". Same thing. No, you did not show this. Your data may "suggest" this.

(7) p.16 line 418: "indicating" co-translational -> "suggesting" co-translational. Same thing.

(8) You cite (24) and (25) for structural work on TRAP. Also, PMID: 36867692 should be cited in the same context.

(9) Fig. 1A and Fig. S13A: You plot ctCCDC47-FLAG over ctSND3-TwinStrep. Wouldn't it be more convincing to plot the opposite ratio, ctSND3-TwinStrep over ctCCDC47-FLAG?

(10) Fig. S14B shows your cryoEM-structure of *C. thermophilum* next to AF3 models of human and yeast. I suggest to add "(AF3 model)" to the labels "*H. sapiens* (AF3 model)" and "*S. cerevisiae* (AF3 model)". To me, there is still a difference between an experimental structure and an AF3 model.

Version 2:

Reviewer comments:

Reviewer #1

(Remarks to the Author)

The authors of this revised manuscript have done an excellent job of addressing concerns of 4 different reviewers (not just my own). This is a very exciting addition to our understanding of protein translocation of IMPs by fungi and exposes the panoply of MPTs in eukaryotes.

(Remarks on code availability)

Reviewer #2

(Remarks to the Author)

The authors have satisfactorily addressed the comments from the previous round.

Minor comment:

Page 9, line 211: Please, do not describe the SND3 model as an "atomic model", since the resolution in the region is certainly not sufficient for such claims.

(Remarks on code availability)

Reviewer #3

(Remarks to the Author)

In their revised manuscript, the authors have fully addressed all my remaining concerns and I congratulate them on this exciting study!

(Remarks on code availability)

Reviewer #4

(Remarks to the Author)

The authors have appropriately addressed my points in the revised version.

(Remarks on code availability)

I checked that the Zenodo archive exists. It lists two Jupyter notebooks as stated. However, I did not download the data.

REVIEWER COMMENTS

First of all, we would like to thank all reviewers for their thoughtful comments and constructive criticism, which have helped to improve the quality of our manuscript.

Reviewer #1 (Remarks to the Author):

This is significant work and moves the field forward, as it reveals not only a previously undocumented fungal ER translocon, but provides many insights into the architecture, likely function and evolution of co-translational multipass translocons that are responsible for integrating multi-TMD containing proteins into the lipid bilayer during their biogenesis. An integrated biochemical, single particle cryogenic-EM and Molecular Dynamics approach is used to analyze a ribosome-associated translocon complex isolated from the thermostable fungi, *Chaetomium thermophilum*. The authors have investigated the partners of SND3 by using a series of pull downs from genetically tagged strains including SND3-twin strep and then with a FLAG tagged partner, CCDC47, which copurifies with SND3 and is a known member of metazoan multipass translocons. They then purified ribosome associated SND3-CCDC47 in sufficient quantities and purity for single particle analysis, to obtain near atomic maps (avg. 2.2 and 2.6 Å resolution) for the large subunit and somewhat lower for the small subunit, which was not analyzed further as this was not the focus of this paper. Typically, regions of ribosome associated components are often imaged at a resolution which gets worse with distance from the primary binding site and this appears to be the case in this specimen. However, the local resolution is good enough for molecular docking of AlphaFold derived models of Sec61 subunits, CCDC47 and SND3 which provides a reasonable (conservative and accurate) description for the major areas of interest highlighted in the text and Figures (but see comments below). The ribosome complexes appear to be in a POST-like conformation with a tRNA in the pe/E-site (although occupancy is hard to assess, and no nascent chain (NC) density was observed in the lsu tunnel).

Many important points arise from this analysis that are addressed clearly and appropriately in this manuscript. Briefly, a probable multipass translocon in fungi is comprised of membrane proteins, Sec61 (alpha,beta,gamma), SND3, CCDC47 and TRAP alpha. The fungal translocon has many structural similarities to a metazoan multipass translocon that also contains Sec61 and CCDC47, along with TMC01 an insertase with some structural similarities to SND3, although the latter has a different fold and thus, SND3 represents a different family of insertases that are distinct from the canonical Oxa1 derived lineage. Hence, the likely role of SND3 has been identified in the context of a ribosome complex. In both translocons, CCDC47 has a domain inserted into the gap between Sec61 and the lsu that would block nascent chain access to the canonical channel. Intriguingly, the Sec61 beta N-terminus is well resolved in the fungal complex and has both a ribosome binding component and also extends into the Sec61 vestibule wherein it may also block NC access. The presence of TRAP alpha (the lone subunit from the vertebrate 4 subunit complex that is present in fungi and presumably contains the active site in all organisms within this complex) is present in a similar but somewhat altered radial position on Sec61 that may position it to work in concert with other components in the multipass translocon.

The arrangement of transmembrane regions in the SND3 multipass translocon creates a lipid filled cavity that is smaller than in the metazoan complex, which could be involved in the integration of membrane protein TMDs, coordinated with SND3s putative role in being able

to accommodate charged/polar regions given its charge distribution within a groove that penetrates partway across the membrane. The possible effects of SND3 on water structure and neighboring phospholipids (via their charged head groups) was evaluated with Molecular Dynamics and are consistent with the authors suggestion that local and partial bilayer disruption/disorder induced by the topology of SND3 may be critical to insertase activity, in the context of a nascent chain with multiple TMs during co-translational translocation.

Given the new structural and biophysical data, which is correlated with other published work, a reasonable and general mechanism is proposed for the fungal MPT that also brings to bear Asterix which was not present in the isolated structure but was shown to interact with SND3-TwinStrep in pulldown experiments; modeling indicates that Asterix would not be sterically hindered in binding to CCDC47, which suggest that it may be a more dynamic component. This work provides a unifying principal for the function of both fungal and metazoan MPTs.

A few technical comments to be addressed in this work:

With regards to map quality:

1. Table 1 : The much larger and better defined ribosome dominates resolution and validation statistics. It would be useful to have a section with local resolution range and validation statistics for ribosome associated components that are important to the focal point of the paper (Sec61 channel, CCDC47 and SND3). Perhaps a Q-factor, or similar metric for model fit into density (FSC0.5 model vs segmented maps) for the Alphafold3 model components and some description of changes in the models when they were "morphed" by refinement into the density (RMSDs).

In the revised manuscript, we have updated **Supplementary Fig. 3** to include the following additional panels: (1) local resolution estimates for the distinct regions of the map corresponding to the individual SND3 translocon components, and (2) superimpositions (including RMSD) between the initial AF3 models and final structural models for each subcomplex described in the text. **Supplementary Figs. 5A, 6A and 6G** show an overview of the final structural models within the cryo-EM density.

2. The local resolution map figures are small and it's hard to judge the overall quality, ie where side chains are clearly resolved for the translocon components. From the shown Figures it seems like side chains would be resolved at the important interfaces described in the text, but a few examples of map quality in these regions would be useful to the reader, along with some description of the overall distribution of side chain densities and the regions where one is able to discern adjacent helical turns (cf. for the docking of TRAP alphas TMD etc. for example).

We have included a new figure (**Supplementary Fig. 4**) which shows map quality within different regions of the structure. The selected regions represent different local resolutions and some biologically relevant interfaces described in the text.

3. There was controversy in the early literature concerning whether the ribosome made a

tight seal to the Sec61-like channel, while the first cryogenic 3D maps showed a sizable gap and suggested that the gap would allow access to this region that might facilitate co-translational folding of cytoplasmic domains of membrane proteins or some other functions. Clearly in this work the gap is used by accessory factors to modulate channel activity and promote integration of multipass membrane proteins that bypass the central pore of the Sec-like channel, while Sec61 alpha/gamma, still function as a ribosome docking site and organizational centerpiece to assemble a larger translocon with enhanced functions. Chaetomium Sec61 alpha-lsu interactions via the 6/7 and 8/9 loops play an important role in forming this translocon and in general, these interactions have not been resolved at fairly high resolution; it would be useful to present some data on this critical interface, which occurs in close proximity and within the tunnel exit.

We thank the reviewer for this suggestion. Indeed, these interactions formed by *ctSEC61 α* are well resolved in our map and are shown in detail as part of a new figure (**Supplementary Fig. 5**).

4. pg 14, line 351 : "sequentially" is not correct, I think they mean "sequence-wise".

We now say "in terms of sequence and structure".

Reviewer #2 (Remarks to the Author):

Here, Yang and co-workers present a break-through structural study of the SND system, probably the least understood ER machinery for membrane protein targeting and insertion in eukaryotic cells. The manuscript starts with a procedure for isolation of the complex (together with the ribosome) followed by structure determination by cryo-EM and molecular dynamics simulations. The membrane-embedded SND component, Snd3, appears in association with the Sec61 translocon and CCDC47 protein, previously described as a part of membrane protein insertase. CCDC47 serves to stabilize the closed conformation of the Sec61 translocon, here being assisted by the cytoplasmic extension of the Sec61-beta subunit. The structural data is employed as an input for MD simulations, which confirm the distortion of the lipid bilayer and high lipid mobility at the interface of Sec61 and Snd3, which indirectly supports the hypothesis that Snd3 is the insertase for membrane proteins, that utilizes similar mechanism as Oxa1/YidC-type insertases.

The results of the study are of great interest, especially in light of the earlier publications on membrane protein insertases from mammalian cells by the groups of RS Hegde and R Keenan. The experimental data are solid and provide a unique, pioneering view on the SND pathway. My major comments are related to the data interpretation and discussion.

1. Differently to Sec61-BOS-GEL-PAT complex trapped in presence of a defined substrate protein, the Sec61-SND3 translocon is purified from the crude ribosome extract. As a result, no nascent protein is seen - neither in the membrane, not at the interface or even within the ribosome exit tunnel. That may be a result of averaging, but can also be interpreted as the insertase complex being in its idle state. The studies on the mammalian system showed that the multipass translocon is assembled in response to the specific nascent protein, so one

should consider such scenario when evaluating the Snd3-Sec61 data. In particular, the Discussion on pages 15-16 sounds rather speculative, as the resolved structure is interpreted solely as an active state. Also the title/abstract should be adjusted, as the presented data (shown in Suppl. Figure 13) are not sufficient to claim the Snd3-Sec61 as a “multipass translocon”.

We think that the stability of the SND3 translocon in the absence of nascent chain could at least in part be attributed to the thermostability of this *C. thermophilum* complex, as seen for other cryo-EM structures (e.g. Kornprobst et al., 2016). However, we acknowledge that in the absence of the nascent chain, the complex can indeed be classed as ‘idle’ and that capturing the SND3 translocon in the presence of substrate should certainly be a goal for future work. We have now made these points at the end of the discussion. Nevertheless, we think the structural and evolutionary relationship between the SND3 translocon and the metazoan multipass translocon is compelling and therefore relevant to discuss in an evolutionary and mechanistic context.

The title has now been changed to “SND3 is the membrane insertase within a distinct SEC61 translocon complex”, whilst use of the word “indicate” in the abstract makes a suggestive reference to the multipass translocon.

2. In connection to the point above, I would recommend to address the visualized complex as Sec61-SND3 translocon/insertase or alike, as both elements are playing distinct roles, and an interplay upon insertion of substrates, e.g. switching between “back-of-Sec61” and Sec61-mediated “lateral gate” routes cannot be excluded.

We completely agree with the reviewer that both the SEC61 lateral gate and SND3 insertase could play a role during substrate insertion. Indeed, we already chose the name “SND3 translocon” to incorporate both the **SND3** insertase and SEC61 **translocon** components. We find this designation memorable due to its simplicity and is in-line with the previous naming of e.g. the “TMCO1 translocon” or “multipass translocon” in the literature, therefore we would prefer to not change the name.

3. Comparison of Snd3 and TMCO1 within the insertase complex does not look convincing, and may be toned down in the discussion (page 14). The proteins occupy quite different positions/ orientations relatively to Sec61, and also the role of TMCO1 within metazoan insertase complex is not clear yet.

Whilst SND3 and TMCO1 have different folds and orientations with respect to the SEC61 translocon, our superimposition of the SND3 translocon and metazoan multipass translocon does show that they overlap significantly in their position:

In addition, both SND3 and TMCO1 have the characteristic features of a membrane insertase, including a membrane embedded hydrophilic groove (indeed, TMCO1 belongs to the Oxa1 superfamily) and a propensity to scramble lipids *in silico* (Li et al., 2024). Although, their insertase activities are yet to be directly shown, TMCO1 is suggestively positioned adjacent to an inserting substrate (Smalinskaite et al., 2022). We therefore find the overlapping positions of SND3 and TMCO1 within their respective translocon complexes relevant to discuss. However, we have now used more suggestive language when discussing this point in the results and discussion section.

4. The architecture of the Snd3 claw domains is quite unusual (clearly different to the helical extension of TMCO1/Oxa1 insertases), and so their position at the membrane interface is. Suppl. Figure 10 suggests certain flexibility within the “claws”, but possibly more detailed explanation of their dynamics may be useful, to learn whether they stay associated with the membrane or detach/displace towards the cytoplasm.

We agree that the architecture of the claw domains is unusual and have further realized when addressing this comment that the previously defined C-claw can be divided into two distinct features i.e. the “C-claw” loop that points towards the membrane (residues 135-173) and a “C-tail”, which comprises a helix that caps the N-claw (residues 173-183) and an unstructured C-terminal tail (residues 184-193) that also points towards the membrane. These definitions have been updated in the text and **Figs. 3A and B**. This is relevant when discussing the flexibility of the *ct*SND3 cytosolic domains, as the RMSF analysis of the atomistic MD simulations (now **Supplementary Fig. 13A**) show that the C-claw is rather stable, whilst the N-claw and C-tail, which together form the hydrophilic arch (previously gate, see response to reviewer 3) are flexible.

In response to the reviewer’s comment, we have now extended our analysis of the MD simulations for these flexible regions. In **Supplementary Fig. 13B**, we overlay 300 simulation snapshots of the N-claw, C-claw and C-tail equally spaced over 300 ns to visualize the flexibility and lipid interactions of these regions. The hydrophobic C-claw loop is stably embedded in the bilayer, which explains its limited mobility in the simulations. In contrast, we see that charged residues in the N-claw (E70, E71) and C-tail (R185, K189, E190, E191) interact with the lipid headgroups (phosphate, choline in POPC, inositol in POPI and ethanolamine in POPE). We analysed this behaviour by computing the time evolution of the minimum distance between the heavy atoms (excluding hydrogen) of the N-claw and C-tail, and the lipid headgroups. The plots presented in **Supplementary Fig. 13C** reflect the stable tethering of both regions with the membrane surface. We have now added these points to the results section.

5. The transmembrane rod-shaped density appearing between Sec61-gamma and Sec61-alpha TMH 8, and so in proximity to the lateral gate, seems to be ignored in the description of the cryo-EM data.

We observed two additional distinct densities around the SEC61 translocon, including the rod-shaped density the reviewer is referring to. We believe their size and shape most likely resembles a stably bound lipid or detergent molecule isolated with the complex, rather than a substrate peptide. Indeed, we found that lipids occupy positions similar to these densities in our atomistic MD simulations, providing further weight to this argument:

We decided not to add this to the revised manuscript as we felt it would interrupt the flow of the text and is not particularly relevant to the story.

6. For presentation of the AlphaFold3 prediction plots, e.g. Suppl. Figure 3D, the areas/regions for individual proteins/subunits should be indicated for clarity.

We have added labels to the AF3 prediction plots, which now appear together in **Supplementary Fig. 3D-F**.

Reviewer #3 (Remarks to the Author):

In their paper SND3 is the membrane insertase within a fungal multipass translocon Yang et al. describe a new function of SND3 as a membrane protein insertase in the thus far ill-characterized SND pathway. This is based on a cryo-EM structure of a SND3:Sec61:CCDC47:TRAP α complex together with MD simulations. Based on these data, the authors suggest the observed complex to correspond to the multipass translocon in higher eukaryotes.

Taken together, this study provides compelling evidence for SND3 being a TMD insertase in the ER membrane of fungi and reveals structural details of a supercomplex it engages in. Also, evolutionary insights into membrane protein biogenesis in different species can be derived from this study. A few points should be addressed before publication:

- A recent study has proposed that SND2 acts as a SRP release factor, a very different function as it seems the authors of this study now propose. This should be discussed in more detail. As of now it remains unclear, how "the recently assigned function of human TMEM208 in accelerating release of substrates from SRP is also likely to be applicable to yeast SND2"

We think there is perhaps some confusion here, as our manuscript deals with the structure and function of SND3, not SND2. In the discussion, we wanted to make the observation that SND3 is not conserved in metazoa (and is likely replaced by TMC01), whilst SND2 is conserved throughout eukaryotes. Actually, we think the proposed role of SND2 as an SRP release factor is highly complementary to our designation of SND3 as a membrane insertase; given that SND2 and SND3 are thought to interact, perhaps there is some handover from SRP to SND3 via SND2.

- What exactly do the authors mean by "it is likely that this SEC61 β arrangement is not unique to *C. thermophilum* and has perhaps not been observed in previous structures due to differences in experimental setup" – which differences are they referring to, as this is indeed an unexpected observation.

We think there are two possibilities. Firstly, the absence of a nascent chain in our structure has somehow allowed the stabilization of the SEC61 β arrangement we observe compared to in the metazoan multipass translocon structure where a nascent chain is present. Secondly, this thermostable complex isolated from *C. thermophilum* has enabled a structure with increased resolution and/or decreased mobility for a flexible element. We find this more likely, as this has been seen before in other *C. thermophilum* cryo-EM structures (e.g. Kornprobst et al., 2016, Agip et al., 2025), and the components of the SND3 translocon are generally better resolved than those in the multipass translocon. We have now expanded this point in the discussion.

- Although claimed, the authors do not formally show SND3 to be an insertase. Any functional data on this would significantly strengthen the impact of this work, but may be beyond the scope of the current study.

We agree and are indeed trying to establish a functional assay for the SND3 translocon, however given the complexity and timescale of these experiments we do think this is beyond the scope of this manuscript.

- The depiction of the MS data in Fig. 1A/B is unusual, normally an enrichment over control and the entire volcano plot should be shown.

We have adjusted the volcano plots in **Fig. 1A** and **Supplementary Fig. 16A** to show enrichment over control with positive enrichment values (our previous plot showed control over enrichment). Furthermore, we want to point out that the plot contains the entire data for the IP experiment: we observed a strong enrichment for the binders and no enrichment for unspecific binders in our control samples, which we attribute to our rigorous washing

protocol. We also did not use any data imputation. All proteomics data are accessible for inspection on the PRIDE repository with the ID PXD060914.

- In Fig. 3 C, what is the designation “gate” based on? Is it mobile? Where different conformations observed?

In this case, we used the term “gate” to refer to an entry rather than a mobile element. Although this region does show some flexibility (**Supplementary Fig. 13**), we agree the nomenclature is misleading. Due to the defined shape visualized in **Fig. 3C**, we now rename this region the hydrophilic “arch”.

- The model in Fig. 5 proposes pairwise TMD insertion; what evidence is this based on?

Our cryo-EM structure and the mutually exclusive distribution of SND3 and TMCO1 across eukaryotes together suggest that the SND3 translocon is the multipass translocon in fungi. As such, we anticipate that its preferred substrates are multipass membrane proteins, with some existing evidence presented in **Supplementary Fig. 16**. As stated in the discussion, the model presented in **Fig. 5B**, including the pairwise TMD insertion, is inferred from the current model for the mechanism of multipass membrane protein insertion in metazoa (Smalinskaite et al., 2022). Here, the authors show that 1) the inserted TMD2-TMD3 pair of a multipass membrane protein intermediate (Rho-4TMD) cross-links to TMCO1 and the back of the SEC61 translocon respectively, 2) insertion of the TMD2-TMD3 pair is reduced when the TMCO1-OPT1 complex is deleted and 3) a TMD pair with a short connecting loop does not require the SEC61 translocon for insertion. A recent cross-linking study also showed that the bacteria Oxa1 superfamily insertase YidC inserts a pair of substrate TMDs (Kaufmann et al., 2025). Altogether, existing evidence in the literature indicates that membrane insertases with a hydrophilic groove, like that found in SND3, are able to translocate short hydrophilic sequences into the ER lumen, which for a multipass membrane protein, reside between pairs of TMDs. However, we acknowledge that the model presented in **Fig. 5B** needs to be rigorously tested in future work on the SND3 translocon.

Reviewer #4 (Remarks to the Author):

This interesting manuscript presents a cryo-EM structure of the membrane insertase SND3 together with ribosome, Sec61, Trap-alpha and CCDC47. To my knowledge, it is the first experimental structure of SND3. The structure also reveals the previously unknown conformation of the N-terminal tail of Sec61beta suggesting that it establishes contacts with the ribosome. Sec61 is observed in a closed state. In the observed conformation, access to the Sec61alpha pore is blocked by CCDC47 and Sec61beta. Trap-alpha is sideways displaced toward SND3 relative to previous cryoEM structures (without SND3).

Also, atomistic and coarse-grained MD simulations reveal that the lipid bilayer is thinned next to SND3 and suggest that SND3 acts as a flippase.

The manuscript is very well written and illustrated, cites relevant previous work, and ties in nicely with the current literature. Several omissions are mentioned below.

Points to be addressed in a revision:

(1) Analysis of the MD simulations (which were understandably conducted without ribosome) is presented not in enough detail. Fig. S10 shows RMSF plots of all subunits. Fig. 3F shows membrane thinning. Movie #2 illustrates mobility of a phospholipid.

(a) I suggest that the authors also comment on the stability of the individual protein subunits (RMSD wrt. to starting structure) and on the stability of the contacts between subunits. The atomistic simulations are likely too short (300 ns) to be affected by the absence of stabilizing ribosome. However, the coarse-grained simulations could show deviations.

We have now analysed the stability of the protein subunits and their interfaces in the MD simulations in more detail. We have computed the RMSD of the translocon protein subunits with respect to their initial configurations from both coarse-grained (**left panel, Supplementary Fig. 12A**) and three atomistic MD simulations (**left panel, Supplementary Fig. 12B and Source Data**). However, for a better understanding of the structural dynamics of the proteins, the RMSD values were calculated by fitting to the coordinates of the atoms that exhibit lower fluctuations (neglecting the more dynamic parts such as disordered regions). Therefore, by considering the heavy atoms which have RMSF values $< 1.5 \text{ \AA}$ (dashed black lines in **middle panel, Supplementary Figs. 12A and 12B**), we have recalculated the RMSD of the proteins (**right panel, Supplementary Figs. 12A and 12B**). These plots show that the membrane-embedded regions of the protein subunits are stable over the course of the simulations, even in the absence of the ribosome. Both the cytosolic domains of SEC61 β and CCDC47 (captured more vividly in atomistic simulations) show significant fluctuations, however this is likely due, at least in part, to the absence of their stabilising interactions with the ribosome that are present in the cryo-EM structure. The flexible cytosolic regions of SND3 are also analysed more thoroughly in terms of flexibility and interactions in the revised manuscript (**Supplementary Fig. 13**), which is discussed in more detail in response to reviewer 2.

To quantify the stability of the interfaces between the different proteins in the SND3 translocon complex, we analysed the number of native contacts between pairs of subunits from the coarse-grained MARTINI3 simulations over a period of 10 μs . According to our definition, a native contact between two residues is formed when the distance between any two heavy atoms is $< 4.5 \text{ \AA}$ in the experimental structure. In the MD simulations, we consider the contacts to remain formed until the minimum distance between them crosses 7 \AA , to filter out transient openings, beyond which the contact is considered broken until it again reached $< 4.5 \text{ \AA}$. **Supplementary Fig. 12C** shows the time evolution of the number of native contacts for the amino-acid pairs which have at least 5 heavy atom contacts in the experimental structure. The number of contacts remain steady throughout the trajectory, which indicates that the interfaces are indeed stable in the simulation, consistent with the results of our visual inspection. We have included these data in the results sections of the revised manuscript.

The three replicates for these atomistic MD simulations (**Supplementary Fig. 12B and Source Data**) have now been used to derive **Supplementary Fig. 11B** (previously obtained from a single simulation of SND3 alone), where we plot the mean phospholipid tilt angle distribution for lipids that are far and near with respect to *ct*SND3. The standard errors are denoted by the shaded regions and, for clarity, we have now dropped the separate distributions for the cytosolic and luminal leaflets. The 'near' lipids are now defined as those

with their phosphorus atom $< 5 \text{ \AA}$ from any heavy atom in SND3, whilst the previously defined distance of $< 3 \text{ \AA}$ included hydrogen atoms and was more susceptible to fluctuations. Altogether, we think these changes make the data more robust and easier to interpret.

(b) p.26 line 721 suggests that coarse-grained simulations were carried out without position restraints in the second part of the CG simulations. How about restraints to stabilize the protein secondary structure elements?

We apologise for the confusion. The “position restraints” in this statement refer to the harmonic restraints which were applied on the protein and lipid atoms (beads) to allow the solvent (water and ions) to relax independent of the protein and lipid molecules. During the equilibration process, these restraints were removed to allow equilibration of the whole system. This sequential equilibration protocol ensured proper relaxation of the system prior to the production simulation.

To stabilise the secondary structure of the protein in the coarse-grained MARTINI3 simulation, separate elastic bonds were applied between protein atoms. This is different from and independent of the aforementioned restraints. This is mentioned in the beginning of the coarse-grained simulations methods section: “Elastic bonds with force constants of $500 \text{ kJ mol}^{-1} \text{ nm}^{-2}$ were applied between the protein atoms within a distance range of 0.5 and 0.9 nm in individual chains to maintain their secondary structures during the simulations.” Note that in the production phase the atomistic MD simulations were performed without restraints.

(2) p.7 line 152: Why is ctSec61alpha “also” present in a closed conformation. Do you also see it in open conformations elsewhere? Or is another subunit also closed?

“Also” here refers to it being the same (closed) conformation observed in the multipass translocon. But we agree it was misleading, and have removed this word from the sentence.

(3) p.9 line 204 “suggesting it could play a similar role in co-translational protein translocation”. Do you suggest here that TRAPalpha affects translocation via SND3? This is later picked up in line 383 in the discussion section - which is the appropriate place for speculation.

This was meant more generally in the context of the TRAP α luminal domain; in metazoa, its role is poorly defined other than being important for co-translational translocation, but mutagenesis studies in *C. elegans* demonstrate the importance of some hydrophobic residues (Jaskolowski et al., 2023) that are partially conserved in *C. thermophilum*. We have now removed this more hypothetical part of the sentence.

(4) p.11 line 262: “our data provide strong evidence for ctSND3 being a membrane insertase”. I don't agree to this. You do not present functional membrane-insertion assays. You connect your structural data to experimental work presented elsewhere. The MD simulations monitor the mobility of a phospholipid. This may resemble a typical SND3 cargo in some aspects, but a phospholipid is certainly not a multipass integral membrane protein. This sentence should be reworded.

We have changed the sentence to “our data altogether suggest ctSND3 is a membrane insertase”.

(5) p.13 line 342 "co-translational insertion ... shown here". Same thing. You do not present data from insertion assays. Hence, you cannot state this here.

We have changed “shown” to “proposed”.

(6) p.16 line 410 "these data altogether show that SND3 inserts TMDs". Same thing. No, you did not show this. Your data may "suggest" this.

We have replaced “show” with “suggest”.

(7) p.16 line 418: "indicating" co-translational -> "suggesting" co-translational. Same thing.

Corrected.

(8) You cite (24) and (25) for structural work on TRAP. Also, PMID: 36867692 should be cited in the same context.

We have now cited this paper when commenting on the general overall similarities between *C. thermophilum* and metazoan TRAP α , as the luminal domains sit in a similar position. However, in this structure the TRAP α TMD occupies a unique position, interacting with the TRAP γ subunit not present in fungi, therefore it is not appropriate to cite this paper when comparing the TMDs.

(9) Fig. 1A and Fig. S13A: You plot ctCCDC47-FLAG over ctSND3-TwinStrep. Wouldn't it be more convincing to plot the opposite ratio, ctSND3-TwinStrep over ctCCDC47-FLAG?

Our labelling of the MS data (ctSND3/CCDC47) actually referred to the genomically modified *C. thermophilum* strain used in the experiments, rather than this ratio. The data instead represent enrichment from this strain over control pull-downs from wild-type *C. thermophilum* with no genomic modifications. We understand why this was confusing, and have removed the label from these figures; we think the experiment is clarified sufficiently in the legend.

(10) Fig. S14B shows your cryoEM-structure of *C. thermophilum* next to AF3 models of human and yeast. I suggest to add "(AF3 model)" to the labels "H. sapiens (AF3 model)" and "S. cerevisiae (AF3 model)". To me, there is still a difference between an experimental structure and an AF3 model.

We agree and had made this distinction in the figure legend and main text. We now explicitly label the AF3 and experimental models within **Supplementary Fig. 17**. We also realized that in the crystal structure of *Thermus thermophilus* SecYEG, the SecG cytoplasmic loop is also occluding the SecY channel, despite a lack of sequence and topology

conservation with eukaryotic SEC61 β . We have therefore added a comparison with this structure to the same figure and made this point in the discussion.